# Fair Node Representation Learning via Adaptive Data Augmentation

## Abstract

Node representation learning has demonstrated its efficacy for various applications on graphs, which leads to increasing attention towards the area. However, fairness is a largely under-explored territory within the field, which may lead to biased results towards underrepresented groups in ensuing tasks. To this end, this work theoretically explains the sources of bias in node representations obtained via Graph Neural Networks (GNNs). Our analysis reveals that both nodal features and graph structure lead to bias in the obtained representations. Building upon the analysis, fairness-aware data augmentation frameworks on nodal features and graph structure are developed to reduce the intrinsic bias. Our analysis and proposed schemes can be readily employed to enhance the fairness of various GNN-based learning mechanisms. Extensive experiments on node classification and link prediction are carried out over real networks in the context of graph contrastive learning. Comparison with multiple benchmarks demonstrates that the proposed augmentation strategies can improve fairness in terms of statistical parity and equal opportunity, while providing comparable utility to state-of-the-art contrastive methods.

## 1 Introduction

Graphs are widely used in modeling and analyzing complex systems such as biological networks or financial markets, which leads to a rise in attention towards various machine learning (ML) tasks over graphs. Specifically, node representation learning is a field with growing popularity. Node representations are mappings from nodes to vector embeddings containing both structural and attributive information. Their applicability on ensuing tasks has enabled various applications such as traffic forecasting (Opolka et al., 2019), and crime forecasting (Jin et al., 2020). Graph neural networks (GNNs) have been prevalently used for representation learning, where node embeddings are created by repeatedly aggregating information from neighbors for both supervised and unsupervised learning tasks (Kipf & Welling, 2017; Veličković et al., 2018; García-Durán & Niepert, 2017).

It has been shown that ML models propagate pre-existing bias in training data, which may lead to discriminative results for ensuing applications (Dwork et al., 2012; Beutel et al., 2017). Particular to ML over graphs, while GNN-based methods achieve state-of-the-art results for graph representation learning, they also amplify already existing biases in training data (Dai & Wang, 2020). For example, nodes in social networks tend to connect to other nodes with similar attributes, leading to denser connectivity between nodes with same sensitive attributes (e.g., gender) (Hofstra et al., 2017). Thus, by aggregating information from the neighbors, the representations obtained by GNNs may be highly correlated with the sensitive attributes. This causes discrimination in ensuing tasks even when the sensitive attributes are not directly used in training (Hajian & Domingo-Ferrer, 2013).

Data augmentation has been widely utilized to improve generalizability in trained models, as well as enable learning in unsupervised methods such as contrastive or self-supervised learning. Augmentation schemes have been extensively studied in vision (Shorten & Khoshgoftaar, 2019; Hjelm et al., 2018) and natural language processing (Zhang et al., 2015; Kafle et al., 2017). However, there are comparatively limited work in the graph domain due to the complex, non-Euclidean structure of graphs. To the best of our knowledge, (Agarwal et al., 2021) is the only study that designs fairness-aware graph data augmentation in the contrastive learning framework to reduce bias.

This study theoretically investigates the sources of bias in GNN-based learning and in turn improves fairness in node representations by employing *fairness-aware* graph data augmentation schemes. Proposed schemes corrupt both input graph topology and nodal features *adaptively*, in order to reduce the corresponding terms in the analysis that lead to bias. Although the proposed schemes are presented over their applications using contrastive learning, the introduced augmentation strategies can be flexibly utilized in several GNN-based learning approaches together with other fairness-enhancement methods. Our contributions in this paper can be summarized as follows:

**c1)** We theoretically analyze the sources of bias that is propagated towards node representations in a GNN-based learning framework.

**c2)** Based on the analysis, we develop novel fairness-aware graph data augmentations that can reduce potential bias in learning node representations. Our approach is adaptive to both input graph and sensitive attributes, and to the best of our knowledge, is the first study that tackles fairness enhancement through *adaptive* graph augmentation design.

**c3)** The proposed strategies incur low additional computation complexity compared to non-adaptive counterparts, and are compatible to operate in conjunction with various GNN-based learning frameworks, including other fairness enhancement methods.

**c4)** Theoretical analysis is provided to corroborate the effectiveness of the proposed feature masking and node sampling augmentation schemes.

**c5)** Performance of the proposed graph data augmentation schemes is evaluated on real networks for both node classification and link prediction tasks. It is shown that compared to state-of-the-art graph contrastive learning methods, the novel augmentation schemes improve fairness metrics while providing comparable utility measures.

## 2 RELATED WORK

**Representation learning on graphs.** Conventional graph representation learning approaches can be summarized under two categories: factorization-based and random walk-based approaches. Factorization-based methods aim to minimize the difference between the inner product of node representations and a deterministic similarity metric between them (Ahmed et al., 2013; Cao et al., 2015; Ou et al., 2016). Random walk-based approaches, on the other hand, employ stochastic measures of similarity between nodes (Perozzi et al., 2014; Grover & Leskovec, 2016; Tang et al., 2015; Chen et al., 2018). GNNs have gained popularity in representation learning, for both supervised (Kipf & Welling, 2017; Veličković et al., 2018; Hu et al., 2019; Wu et al., 2019), and unsupervised tasks, e.g., (García-Durán & Niepert, 2017; Hamilton et al., 2017). Specifically, recent success of contrastive learning on visual representation learning (Wu et al., 2018; Ye et al., 2019; Ji et al., 2019) has paved the way for contrastive learning for unsupervised graph representation learning.

**Graph data augmentation.** Augmentation strategies have been extensively investigated in vision (Shorten & Khoshgoftaar, 2019; Hjelm et al., 2018) and natural language processing (Zhang et al., 2015; Kafle et al., 2017) domains. However, the area is comparatively under-explored in the graph domain due to the complex, non-Euclidean topology of graphs. Graph augmentation based on graph structure modification has been developed to improve the utility of ensuing tasks (Rong et al., 2019; Zhao et al., 2020; Chen et al., 2020a). Meanwhile, graph data augmentation has been used to generate graph views for unsupervised graph contrastive learning, see, e.g., (Veličković et al., 2019; Opolka et al., 2019; Zhu et al., 2020; 2021), which achieves state-of-the-art results in various learning tasks over graphs such as node classification, regression, and link prediction (Opolka et al., 2019; Veličković et al., 2019; You et al., 2020; Zhu et al., 2020; Peng et al., 2020; Hassani & Khasahmadi, 2020). Among which, (Zhu et al., 2020) is the first study that aims to maximize the agreement of node-level embeddings across two corrupted graph views. Building upon (Zhu et al., 2020), (Zhu et al., 2021) develops adaptive augmentation schemes with respect to various node centrality measures and achieves better results. However, none of these studies are fairness-aware.

**Fairness-aware learning on graphs.** A pioneering study tackling the fairness problem in graph representation learning based on random walks is developed in (Rahman et al., 2019). In addition, adversarial regularization is employed to account for fairness of node representations (Dai & Wang, 2020; Bose & Hamilton, 2019; Fisher et al., 2020) where (Dai & Wang, 2020) is presented specifically for node classification, and (Fisher et al., 2020) works on knowledge graphs. (Buyl & De Bie, 2020) also aims to create fair node representations by utilizing a Bayesian approach where sensitive information is modeled in the prior distribution. Contrary to these aforementioned works,

our framework is built on a theoretical analysis (developed within this paper). Similar to the works mentioned above, the proposed methods can be utilized within the learning process to mitigate bias by modifying the learned model (i.e., in-processing fairness strategy). In addition, the proposed schemes can also be regarded as "pre-processing" tools, implying their compatibility to a wide array of GNN-based learning schemes in a versatile manner. Furthermore, (Ma et al., 2021) carries out a PAC-Bayesian analysis and connects the concept of subgroup generalization to accuracy disparity, and (Zeng et al., 2021) introduces several methods including GNN-based ones to decrease the bias for the representations of heterogeneous information networks. While (Li et al., 2021; Laclau et al., 2021) modify adjacency to improve different fairness measures specifically for link prediction, (Buyl & De Bie, 2021) designs a regularizer for the same purpose. With a specific interest on individual fairness over graphs, (Dong et al., 2021) employs a ranking-based strategy. A biased edge dropout scheme is proposed in (Spinelli et al., 2021) to improve fairness. However, the scheme therein is not adaptive to the graph structure (the parameters of the framework are independent of the input graph topology). Fairness-aware graph contrastive learning is first studied in (Agarwal et al., 2021), where a layer-wise weight normalization scheme along with graph augmentations is introduced. However, the fairness-aware augmentation utilized therein is designed primarily for counterfactual fairness.

## 3 FAIRNESS IN GNN-BASED REPRESENTATION LEARNING

GNN-based approaches are the state-of-the-art for node representation learning. However, it has been demonstrated that the utilization of graph structure in the learning process not just propagates but also amplifies a possible bias towards certain sensitive groups (Dai & Wang, 2020). To this end, this section investigates the sources of bias in the generated representations via GNN-based learning. It carries out an analysis revealing that both nodal features and graph structure lead to bias, for which several graph data augmentation frameworks are introduced.

### 3.1 PRELIMINARIES

This study aims to learn fairness-aware nodal representations for a given graph $\mathcal{G} := (\mathcal{V}, \mathcal{E})$ where $\mathcal{V} := \{v_1, v_1, \cdots, v_N\}$ denotes the node set, and $\mathcal{E} \subseteq \mathcal{V} \times \mathcal{V}$ represents the edge set. $\mathbf{X} \in \mathbb{R}^{N \times F}$ and $\mathbf{A} \in \{0,1\}^{N \times N}$ are used to denote the feature and adjacency matrices, respectively, with the $(i,j)$-th entry $a_{ij} = 1$ if and only if $e_{ij} := (v_i, v_j) \in \mathcal{E}$. Degree matrix $\mathbf{D} \in \mathbb{R}^{N \times N}$ is defined to be a diagonal matrix with the $n$th diagonal entry $d_n$ denoting the degree of $v_n$. For the fairness examination, sensitive attributes of the nodes are represented with $\mathbf{s} \in \{0,1\}^{N \times 1}$, where the existence of a single, binary sensitive attribute is considered. In this work, unsupervised methods are chosen as enabling schemes for the representation generation where given the inputs $\mathbf{A}, \mathbf{X}$, and $\mathbf{s}$, the main purpose is to learn a mapping $f : \mathbb{R}^{N \times N} \times \mathbb{R}^{N \times F} \times \mathbb{R}^{N \times 1} \to \mathbb{R}^{N \times F^L}$ that generates $F^L$ dimensional (generally $F^L \ll F$) unbiased nodal representations $\mathbf{H}^L = f(\mathbf{A}, \mathbf{X}, \mathbf{s}) \in \mathbb{R}^{N \times F^L}$ through an $L$-layer GNN, which can be used in an ensuing task such as node classification. $\mathbf{x}_i \in \mathbb{R}^F$, $\mathbf{h}_i^l \in \mathbb{R}^{F^l}$, and $s_i \in \{0,1\}$ denote the feature vector, representation at layer $l$ and the sensitive attribute of node $v_i$. Furthermore, $\mathcal{S}_0$ and $\mathcal{S}_1$ denote the set of nodes whose sensitive attributes are 0 and 1, respectively. Define inter-edge set $\mathcal{E}^\chi := \{e_{ij} | v_i \in \mathcal{S}_a, v_j \in \mathcal{S}_b, a \neq b\}$, while intra-edge set is defined as $\mathcal{E}^\omega := \{e_{ij} | v_i \in \mathcal{S}_a, v_j \in \mathcal{S}_b, a = b\}$ . Similarly, the set of nodes having at least one inter edge is denoted by $\mathcal{S}^\chi$, while $\mathcal{S}^\omega$ defines the set of nodes that have no inter-edges. The intersection of the sets $\mathcal{S}_0, \mathcal{S}^\chi$ is denoted as $\mathcal{S}_0^\chi$. Additionally, $d_i^\chi$ and $d_i^\omega$ denote the numbers of inter-edges and intra-edges adjacent to $v_i$, respectively. Finally, $|\cdot|$ denotes the entry-wise absolute value for scalar or vector inputs, while it is used for the cardinality when the input is a set.

### 3.2 ANALYSIS FOR BIAS IN GNN REPRESENTATIONS

This subsection presents an analysis to find out the sources of bias in node representations generated by GNNs. Analysis is developed for the mean aggregation scheme in which aggregated representations at layer $l$, $\mathbf{Z}^l \in \mathbb{R}^{N \times F^l}$, are generated such that $\mathbf{z}_i^l = \frac{1}{d_i} \sum_{v_j \in \mathcal{N}(i)} \mathbf{h}_j^{l-1}$ for $i = \{1, \cdots, N\}$, where $\mathbf{z}_i^l$ is the $i$th row of $\mathbf{Z}^l$ corresponding to node $v_i$, $d_i$ denotes the degree of node $v_i$, $\mathcal{N}(i)$ refers to the neighbor set of node $v_i$ (including itself). The recursive relation in a GNN layer in which left normalization is applied for feature smoothing is $\mathbf{H}^l = \sigma(\mathbf{D}^{-1}(\mathbf{A} + \mathbf{I}_N)\mathbf{H}^{l-1}\mathbf{W}^l)$ where $\mathbf{W}^l$ is the weight matrix in layer $l$, and $\mathbf{I}_N \in \mathbb{R}^{N \times N}$ denotes an identity matrix. With these definitions, the

relation between the aggregated information $\mathbf{Z}^l$ and node representations $\mathbf{H}^l$ becomes equivalent to $\mathbf{H}^l = \sigma(\mathbf{Z}^l \mathbf{W}^l)$ at the $l$th GNN layer. As the provided analysis is applicable to every layer, $l$ superscript is dropped in the following to keep the notation simple.

It has been demonstrated that features that are correlated with the sensitive attribute result in bias even when the sensitive attribute is not utilized in the learning process (Hajian & Domingo-Ferrer, 2013). This work provides an analysis on the correlation of $\mathbf{s}$ with $\mathbf{Z}$, and aims to reduce it. Note that, the reduction of correlation can still allow the generation of discriminable representations for different class labels, if the discriminability is provided by non-sensitive attributes. The (sample) correlation between the sensitive attributes $\mathbf{s}$ and aggregated representations can be written as

$$\rho_i = \mathrm{Corr}(\tilde{\mathbf{z}}_{:,i}, \mathbf{s}) = \frac{\sum_{j=1}^N (z_{ji} - \frac{1}{N}\sum_{k=1}^N z_{ki})(\mathbf{s}_j - \frac{1}{N}\sum_{k=1}^N \mathbf{s}_k)}{\sqrt{\sum_{j=1}^N (z_{ji} - \frac{1}{N}\sum_{k=1}^N z_{ki})^2}\sqrt{\sum_{j=1}^N (\mathbf{s}_j - \frac{1}{N}\sum_{k=1}^N \mathbf{s}_k)^2}}$$

where $\tilde{\mathbf{z}}_{:,i}, i = 1 \cdots F$ is the $i$th column of $\mathbf{Z}$. In the analysis, following assumptions are made:
**A1:** Node representations have sample means $\boldsymbol{\mu}_0$ and $\boldsymbol{\mu}_1$ respectively across each group, where $\boldsymbol{\mu}_i = \mathrm{mean}(\mathbf{h}_j \mid v_j \in \mathcal{S}_i)$. Throughout the paper, $\mathrm{mean}(\cdot)$ denotes the sample mean operation.
**A2:** Node representations have finite maximal deviations $\Delta_0$ and $\Delta_1$: That is, $\|\mathbf{h}_j - \boldsymbol{\mu}_i\|_\infty \leq \Delta_i$, $\forall v_j \in S_i$ with $i \in \{0, 1\}$.
Based on these assumptions, the following theorem shows that $\|\boldsymbol{\rho}\|_1$ can be bounded from above, which will serve as a guideline to design a fairness-aware graph data augmentation scheme.

**Theorem 1.** *The total correlation between the sensitive attributes* $\mathbf{s}$ *and representations* $\mathbf{Z}$ *that are obtained after a mean aggregation over graph* $\mathcal{G}$, $\|\boldsymbol{\rho}\|_1$, *can be bounded above by*

$$\|\boldsymbol{\rho}\|_1 \leq \|\mathbf{c}\|_1 (\|\boldsymbol{\delta}\|_1 \max(\gamma_1, \gamma_2) + 2N\Delta) \tag{1}$$

*where* $c_i := \frac{\sqrt{|\mathcal{S}_0||\mathcal{S}_1|}}{N \sigma_{\tilde{\mathbf{z}}_{:,i}}}$, *with* $\sigma_{\boldsymbol{\theta}} := \sqrt{\frac{1}{N}\sum_{n=1}^N (\theta_n - \frac{1}{N}\sum_{i=1}^N \theta_i)^2}, \forall \boldsymbol{\theta} \in \mathbb{R}^N$, $\boldsymbol{\delta} := \boldsymbol{\mu}_0 - \boldsymbol{\mu}_1$, $\gamma_1 := \left|1 - \frac{|\mathcal{S}_0^\chi|}{|\mathcal{S}_0|} - \frac{|\mathcal{S}_1^\chi|}{|\mathcal{S}_1|}\right|$, $\gamma_2 = \left|1 - 2\min\left(\mathrm{mean}\left(\frac{d_m^\chi}{d_m^\chi + d_m^\omega} | v_m \in \mathcal{S}_0\right), \mathrm{mean}\left(\frac{d_n^\chi}{d_n^\chi + d_n^\omega} | v_n \in \mathcal{S}_1\right)\right)\right|$, $\Delta = \max(\Delta_0, \Delta_1)$.

The proof is given in Appendix A.1. The upper bound in equation 1 can be lowered by i) utilizing feature masking which has an effect on the term $\|\boldsymbol{\mu}_0 - \boldsymbol{\mu}_1\|_1$ at the first layer, ii) node sampling that can change the value of $\gamma_1$, iii) edge augmentations that can reduce the value of $\gamma_2$.

## 3.3 Fair Graph Data Augmentations

Data augmentation has been studied extensively in order to enable certain unsupervised learning schemes such as contrastive learning, self-supervised learning or as a general framework to improve the generalizability of the trained models over unseen data. However, the design of graph data augmentations is still a developing research area due to the challenges introduced by complex, non-Euclidean graph structure. Several augmentation schemes over the graph structure are proposed in order to enhance the generalizability of GNNs (Rong et al., 2019; Zhao et al., 2020), while both topological (e.g., edge/node deletion) and attributive (e.g., feature shuffling/masking) corruption schemes have been developed in the context of contrastive learning (Veličković et al., 2019; You et al., 2020; Zhu et al., 2021; 2020). However, none of these works are fairness-aware. Hence, in this work, novel data augmentation schemes that are *adaptive* to the sensitive attributes, as well as the input graph structure are introduced with Theorem 1 as guidelines.

### 3.3.1 Feature Masking

In this subsection, an augmentation framework on nodal features $\mathbf{H}^0 = \mathbf{X}$ is presented in order to mitigate possible intrinsic bias propagated by them. Note that $\|\boldsymbol{\delta}\|_1$ in equation 1 is minimized when all nodal features are the same (i.e., all nodal features masked/zeroed out). However, this would result in the loss of all information in nodal features. Motivated by this, the proposed scheme aims to improve uniform feature masking in terms of reducing $\|\boldsymbol{\delta}\|_1$ for a given *masking budget* (a total amount of nodal features to be masked in expectation). Specifically, the random feature masking scheme used in (You et al., 2020; Zhu et al., 2020) where each feature has the same masking probability is modified to assign higher masking probabilities to the features varying more across

different sensitive groups. Thus, masking probabilities are generated based on the term $|\boldsymbol{\delta}|$. Let $\bar{\boldsymbol{\delta}} := \frac{|\boldsymbol{\delta}| - \min(|\boldsymbol{\delta}|)}{\max(|\boldsymbol{\delta}|) - \min(|\boldsymbol{\delta}|)}$ denote the normalized $|\boldsymbol{\delta}|$, the feature masking probability can then be designed as

$$\mathbf{p}^{(m)} = \min\left(\frac{\alpha \bar{\boldsymbol{\delta}}}{\frac{1}{F}\sum_{i=1}^{F} \bar{\delta}_i}, 1\right) \tag{2}$$

where $\alpha$ is a hyperparameter. The feature mask $\mathbf{m} \in \{0, 1\}^F$ is then generated as a random binary vector, with the $i$-th entry of $\mathbf{m}$ drawn independently from Bernoulli distribution with $p_i = (1 - p_i^{(m)})$ for $i = 1, \ldots, F$. The augmented feature matrix is obtained via

$$\tilde{\mathbf{X}} = [\mathbf{m} \circ \mathbf{x}_1; \ldots; \mathbf{m} \circ \mathbf{x}_N]^\top, \tag{3}$$

where $[\cdot; \cdot]$ is the concatenation operator, and $\circ$ is the Hadamard product. Since the proposed feature masking scheme is probabilistic in nature, the resulting $\tilde{\boldsymbol{\delta}}$ is a random vector with entry $\tilde{\delta}_i$ having

$$|\tilde{\delta}_i| = \begin{cases} |\delta_i|, & \text{with probability } p_i, \\ 0, & \text{with probability } 1 - p_i \end{cases} \tag{4}$$

where $p_i := 1 - p_i^{(m)}$ is the probability that the $i^{th}$ feature is not masked in the graph view. The following proposition shows that the novel, adaptive feature masking approach can decrease $\|\boldsymbol{\rho}\|_1$ compared to random feature masking, the proof of which can be found in Appendix A.2.

**Proposition 1.** *In expectation, the proposed adaptive feature masking scheme results in a lower $\|\tilde{\boldsymbol{\delta}}\|_1$ value compared to uniform feature masking, meaning*

$$E_{\mathbf{p}}[\|\tilde{\boldsymbol{\delta}}\|_1] \leq E_{\mathbf{q}}[\|\tilde{\boldsymbol{\delta}}\|_1] \tag{5}$$

*where $\mathbf{q}$ corresponds to uniform masking with masking probability $1 - q_i$, and $q_i = \frac{1}{F}\sum_{j=1}^{F} p_j$.*

### 3.3.2 Node Sampling

In this subsection, an adaptive node sampling framework is introduced to decrease the term $\gamma_1 := \left|1 - \left(\frac{|\mathcal{S}_0^\chi|}{|\mathcal{S}_0|} + \frac{|\mathcal{S}_1^\chi|}{|\mathcal{S}_1|}\right)\right| = \left|1 - \left(\frac{|\mathcal{S}_0^\chi|}{|\mathcal{S}_0^\chi| + |\mathcal{S}_0^\omega|} + \frac{|\mathcal{S}_1^\chi|}{|\mathcal{S}_1^\chi| + |\mathcal{S}_1^\omega|}\right)\right|$ in equation 1 of Theorem 1, and hence to reduce the intrinsic bias that the graph topology can create. A small $\gamma_1$ suggests a more balanced population distribution with respect to $|\mathcal{S}^\chi|$ and $|\mathcal{S}^\omega|$. Specifically, a subset of nodes is selected at every epoch and the training is carried over the subgraph induced by the sampled nodes. This augmentation mainly aims at reducing the bias by selecting a subset of more balanced groups, meanwhile it also helps reduce the computational and memory complexity in training.

The proposed node sampling is adaptive to the input graph, that is, it depends on the cardinalities of the sets $\mathcal{S}_0^\chi, \mathcal{S}_1^\chi, \mathcal{S}_0^\omega,$ and $\mathcal{S}_1^\omega$. The developed scheme copes with the case $|\mathcal{S}^\chi| \leq |\mathcal{S}^\omega|$. In algorithm design, it is assumed that if $|\mathcal{S}^\chi| \leq |\mathcal{S}^\omega|$ then $|\mathcal{S}_0^\chi| \leq |\mathcal{S}_0^\omega|$ and $|\mathcal{S}_1^\chi| \leq |\mathcal{S}_1^\omega|$ (same for $\mathcal{S}^\omega$), which holds for all real graphs in our experiments, but our design principles can be readily extended to different settings as well.

Given input graph $\mathcal{G}$, the augmented graph $\tilde{\mathcal{G}}$ can be obtained as an induced subgraph from a subset of nodes $\tilde{\mathcal{V}}$. All nodes in $\mathcal{S}^\chi$ are retained, ($\tilde{\mathcal{S}}_0^\chi = \mathcal{S}_0^\chi$ and $\tilde{\mathcal{S}}_1^\chi = \mathcal{S}_1^\chi$), while subsets of nodes $\bar{\mathcal{S}}_0^\omega$ and $\bar{\mathcal{S}}_1^\omega$ are randomly sampled from $\mathcal{S}_0^\omega$ and $\mathcal{S}_1^\omega$ with sample sizes $|\bar{\mathcal{S}}_0^\omega| = |\mathcal{S}_0^\chi|$ and $|\bar{\mathcal{S}}_1^\omega| = |\mathcal{S}_1^\chi|$, respectively . See also Algorithm 1 in Appendix A.3. The cardinalities of node sets in the resulting graph augmentation $\tilde{\mathcal{G}}$ satisfy $\gamma_1 = 0$. (See Appendix A.3 for details.)

**Remark 1.** Note that the resulting graph $\tilde{\mathcal{G}}$ yields $\gamma_1 = 0$ as long as $|\tilde{\mathcal{S}}_0^\chi|/|\tilde{\mathcal{S}}_0| = 1/2 + \phi$ and $|\tilde{\mathcal{S}}_1^\chi|/|\tilde{\mathcal{S}}_1| = 1/2 - \phi$ is satisfied for any $-1/2 < \phi < 1/2$. The presented scheme here simply sets $\phi = 0$, which results in a balanced ratio across groups, but the performance can be improved if $\phi$ is selected carefully for specific datasets.

### 3.3.3 Augmentation on Graph Connectivity

Minimizing $\gamma_2$ to zero in Theorem 1 suggests a graph topology where all nodes in the network have the same number of neighbors from each sensitive group, i.e., $d_i^\omega = d_i^\chi, \forall v_i \in \mathcal{V}$. Since, for this

scenario, $\gamma_2 = \left| 1 - 2\min\left(\text{mean}\left(\frac{d_m^\chi}{d_m^\chi + d_m^\omega} | v_m \in \mathcal{S}_0\right), \text{mean}\left(\frac{d_n^\chi}{d_n^\chi + d_n^\omega} | v_n \in \mathcal{S}_1\right)\right)\right| = 0$. Note that this finding is parallel to the main design idea of Fairwalk (Rahman et al., 2019) in which the transition probabilities are equalized for different sensitive groups in random walks in order to reduce bias in random walk-based representations.

This finding suggests that an ideal augmented graph $\tilde{\mathcal{G}}$ could be generated by deleting edges from or adding edges to $\mathcal{G}$ such that each node has exactly the same number of neighbors from each sensitive group. However, such a per-node sampling scheme is computationally complex and may not be ideal for large-scale graphs. To this end, we propose global probabilistic edge augmentation schemes such that in the augmented graph,

$$\mathbb{E}[|\tilde{\mathcal{E}}^\omega|] = \mathbb{E}[|\tilde{\mathcal{E}}^\chi|]. \tag{6}$$

Here the expectation is taken with respect to the randomness in the augmentation design. It is shown in our experiments that the global approach can indeed help to reduce the value of $\gamma_2$ (see Appendix A.8). Note that even though the strategy is presented for the case where $|\mathcal{E}^\chi| \leq |\mathcal{E}^\omega|$ (which holds for all datasets considered herein), the scheme can be easily generalized to the case where $|\mathcal{E}^\chi| > |\mathcal{E}^\omega|$.

In social networks, users connect to other users that are similar to themselves with higher probabilities (Mislove et al., 2010), hence the graph connectivity naturally inherits bias towards potential minority groups. Motivated by the reduction of $\gamma_2$, the present subsection introduces augmentation schemes over edges to provide a balanced graph structure that can mitigate such bias.

**Adaptive Edge Deletion.** To obtain a balanced graph structure, we first develop an adaptive edge deletion scheme where edges are removed with certain deletion probabilities. Based on the graph structure and sensitive attributes, the probabilities are assigned as

$$p^{(e)}(e_{ij}) = \begin{cases} 1 - \pi, & \text{if } s_i \neq s_j \\ 1 - \frac{|\mathcal{E}^\chi|}{2|\mathcal{E}^\omega_{\mathcal{S}_0}|}\pi, & \text{if } s_i = s_j = 0 \\ 1 - \frac{|\mathcal{E}^\chi|}{2|\mathcal{E}^\omega_{\mathcal{S}_1}|}\pi, & \text{if } s_i = s_j = 1 \end{cases} \tag{7}$$

where $p^{(e)}(e_{ij})$ is the removal probability of the edge connecting nodes $v_i$ and $v_j$, $\pi$ is a hyperparameter, and $\mathcal{E}^\omega_{\mathcal{S}_i}$ denotes the set of intra-edges connecting the nodes in $S_i$. Note that $\pi$ is chosen to be 1 in this work, but it can be selected by schemes such as grid search to improve performance. While this graph-level edge deletion scheme does not not directly minimize $\gamma_2$, it provides a balanced global structure such that $\mathbb{E}_{\mathbf{p}^{(e)}}[|\tilde{\mathcal{E}}^\omega_{\mathcal{S}_0}|] = \mathbb{E}_{\mathbf{p}^{(e)}}[|\tilde{\mathcal{E}}^\omega_{\mathcal{S}_1}|] = \frac{1}{2}\mathbb{E}_{\mathbf{p}^{(e)}}[|\tilde{\mathcal{E}}^\chi|]$, henceforth equation 6 holds in the augmented graph $\tilde{\mathcal{G}}$, see Appendix A.4 for more details.

**Adaptive Edge Addition.** For graphs that are very sparse, edge deletion may not be an ideal graph augmentation as it may lead to unstable results. In this case, an adaptive edge addition framework is developed to obtain a more balanced graph structure. Specifically, for graphs where $|\mathcal{E}^\chi| \leq |\mathcal{E}^\omega|$ holds, $|\mathcal{E}^\omega| - |\mathcal{E}^\chi|$ pairs of the nodes are sampled uniformly from $\mathcal{S}_0^\chi$ and $\mathcal{S}_1^\chi$ with replacement. Then, a new edge is created to connect each sampled pair of nodes, in order to obtain an augmented graph $\tilde{\mathcal{G}}$ for which equation 6 holds. Note that experimental results in Section 4 also show that edge addition may become a better alternative over edge deletion for graphs that are sparse in inter-edges.

**Remark 2.** Overall, while a subset of these augmentation schemes can be employed based on the input graph properties (sparse/dense, large/small), all schemes can also be employed on the input graph sequentially. The framework where node sampling, edge deletion, edge addition are employed sequentially together with feature masking is called 'FairAug'. It is worth emphasizing that edge augmentation schemes should always follow node sampling, as performing node sampling changes the distribution of edges. Since the cardinalities of different sets will be calculated only once (in pre-processing), we note that the proposed augmentations will not incur significant additional cost.

## 4 EXPERIMENTS

In this section, experiments are carried out on 7 real-world datasets for node classification and link prediction tasks. Performances of our proposed adaptive augmentations are compared with baseline schemes in terms of utility and fairness metrics.

## 4.1 Datasets and Settings

**Datasets.** Experiments are conducted on real-world social and citation networks consisting of Pokec-z, Pokec-n (Dai & Wang, 2020), UCSD34, Berkeley13 (Red et al., 2011), Cora, Citeseer, Pubmed. Pokec-z and Pokec-n sampled from a larger social network, Pokec (Takac & Zabovsky, 2012), are used in node classification experiments where Pokec is a Facebook-like, social network used in Slovakia. While the original network includes millions of users, Pokec-z and Pokec-n are generated by collecting the information of users from two major regions (Dai & Wang, 2020). The region information is treated as the sensitive attribute, while the working field of the users is the label to be predicted in classification. Both attributes are binarized, see also (Dai & Wang, 2020). UCSD34 and Berkeley13 are Facebook networks where edges are created based on the friendship information in social media. Each user (node) has 7 dimensional nodal features including student/faculty status, gender, major, etc. Gender is utilized as the sensitive attribute in UCSD34 and Berkeley13. Cora, Citeseer, and Pubmed are citation networks that consider articles as nodes and descriptions of articles as their nodal attributes. In these datasets, the category of the articles is used as the sensitive attribute. Statistics for datasets are presented in Tables 5 and 6 of Appendix A.5.

**Evaluation Metrics.** Performance of the node classification task is evaluated in terms of accuracy. Two quantitative group fairness metrics are used to assess the effectiveness of fairness aware strategies in terms of **statistical parity**: $\Delta_{SP} = |P(\hat{y} = 1 \mid s = 0) - P(\hat{y} = 1 \mid s = 1)|$ and **equal opportunity**: $\Delta_{EO} = |P(\hat{y} = 1 \mid y = 1, s = 0) - P(\hat{y} = 1 \mid y = 1, s = 1)|$, where $y$ and $\hat{y}$ denote the ground-truth and the predicted labels, respectively. Lower values for $\Delta_{SP}$ and $\Delta_{EO}$ imply better fairness performance (Dai & Wang, 2020). For the link prediction task, both accuracy and Area Under the Curve (AUC) are employed as utility metrics. As the fairness metrics, the definitions of statistical parity and equal opportunity are modified for link prediction such that $\Delta_{SP} = |P(\hat{y} = 1 \mid e \in \mathcal{E}^\chi) - P(\hat{y} = 1 \mid e \in \mathcal{E}^\omega)|$ and $\Delta_{EO} = |P(\hat{y} = 1 \mid y = 1, e \in \mathcal{E}^\chi) - P(\hat{y} = 1 \mid y = 1, e \in \mathcal{E}^\omega)|$, where $e$ denotes the edges and $\hat{y}$ is the decision for whether the edge exists.

**Implementation details.** The proposed augmentation scheme is tested on social networks with node representations generated through an unsupervised contrastive learning framework. Node classification and link prediction are employed as ensuing tasks that evaluate the performances of generated node representations. In addition, we also provide link prediction results obtained via an end-to-end graph convolutional network (GCN) model on citation networks. Further details on the implementation of the experiments are provided in Appendix A.7. Contrastive learning is utilized to demonstrate the effects of the proposed augmentation schemes, as augmentations are inherently utilized in its original design (You et al., 2020). Specifically, GRACE (Zhu et al., 2020) is employed as our baseline framework. GRACE constructs two different graph views using random, non-adaptive augmentation schemes, which we replaced by augmentations obtained via FairAug in the experiments. For more details on the employed contrastive learning framework, see Appendix A.6.

**Baselines.** We present the performances on a total of 7 baseline studies. As we examine our proposed augmentations in the context of contrastive learning, graph contrastive learning schemes are employed as the natural baselines. Said schemes are Deep Graph Infomax (DGI) (Veličković et al., 2019), Deep Graph Contrastive Representation Learning (GRACE) (Zhu et al., 2020), Graph Contrastive Learning with Adaptive Augmentations (GCA) (Zhu et al., 2021), and Fair and Stable Graph Representation Learning (NIFTY) (Agarwal et al., 2021). We note that the objective function of NIFTY (Agarwal et al., 2021) consists of both supervised and unsupervised components, and its results for the unsupervised setting are provided here given the scope of this paper. In addition, as another family of unsupervised approaches, random walk-based methods for unsupervised node representation generation are also considered. Such schemes include DeepWalk (Perozzi et al., 2014), Node2Vec (Grover & Leskovec, 2016), and FairWalk (Rahman et al., 2019). Lastly, for end-to-end link prediction, we present results for the random edge dropout scheme (Rong et al., 2019).

## 4.2 Experimental Results

The comparison between baselines and our proposed framework FairAug is presented in Table 1. Note that 'FairAug wo ED' in Table 1 refers to FairAug where edge deletion (ED) is removed from the chain, but all node sampling (NS), edge addition (EA), and feature masking (FM) are still employed. Firstly, the results of Table 1 show that FairAug provides roughly 45% reduction in fairness metrics over GRACE, the strategy it is built upon, while providing similar accuracy

Table 1: Comparative Results with Baselines on Node Classification

| | Pokec-z | | | Pokec-n | | |
| --- | --- | --- | --- | --- | --- | --- |
| | Accuracy (%) | $\Delta_{SP}$ (%) | $\Delta_{EO}$ (%) | Accuracy (%) | $\Delta_{SP}$ (%) | $\Delta_{EO}$(%) |
| DeepWalk | $60.82 \pm 0.77$ | $9.79 \pm 3.43$ | $8.51 \pm 1.67$ | $59.66 \pm 1.02$ | $20.19 \pm 3.38$ | $22.96 \pm 3.44$ |
| Node2Vec | $61.56 \pm 0.50$ | $16.18 \pm 12.40$ | $14.88 \pm 12.31$ | $60.31 \pm 0.75$ | $20.76 \pm 1.64$ | $20.32 \pm 1.36$ |
| FairWalk | $57.48 \pm 0.11$ | $12.03 \pm 10.19$ | $10.95 \pm 8.82$ | $57.44 \pm 0.73$ | $15.00 \pm 1.40$ | $15.72 \pm 1.80$ |
| DGI | $65.56 \pm 1.29$ | $4.82 \pm 1.89$ | $5.81 \pm 0.97$ | $65.71 \pm 0.24$ | $5.18 \pm 2.15$ | $7.16 \pm 2.44$ |
| GRACE | $\mathbf{67.09} \pm 0.47$ | $6.20 \pm 2.22$ | $6.18 \pm 2.46$ | $\mathbf{67.90} \pm 0.13$ | $8.85 \pm 1.39$ | $10.13 \pm 2.11$ |
| GCA | $66.85 \pm 0.71$ | $7.40 \pm 4.14$ | $7.46 \pm 4.09$ | $67.02 \pm 0.43$ | $5.31 \pm 0.62$ | $7.75 \pm 1.49$ |
| NIFTY | $66.09 \pm 0.40$ | $5.86 \pm 1.97$ | $6.19 \pm 1.72$ | $65.44 \pm 0.17$ | $5.18 \pm 2.80$ | $5.78 \pm 3.18$ |
| FairAug | $67.04 \pm 0.69$ | $3.29 \pm 1.66$ | $\mathbf{3.04} \pm 1.37$ | $67.88 \pm 0.45$ | $4.81 \pm 0.59$ | $6.52 \pm 0.99$ |
| FairAug wo ED | $67.38 \pm 0.31$ | $\mathbf{2.66} \pm 3.22$ | $4.26 \pm 2.92$ | $67.57 \pm 0.18$ | $\mathbf{3.37} \pm 0.62$ | $\mathbf{5.13} \pm 1.12$ |

Table 2: Ablation Study on Node Classification

| | Pokec-z | | | Pokec-n | | |
| --- | --- | --- | --- | --- | --- | --- |
| | Accuracy (%) | $\Delta_{SP}$ (%) | $\Delta_{EO}$ (%) | Accuracy (%) | $\Delta_{SP}$ (%) | $\Delta_{EO}$(%) |
| GRACE | $67.09 \pm 0.47$ | $6.20 \pm 2.22$ | $6.18 \pm 2.46$ | $67.90 \pm 0.13$ | $8.85 \pm 1.39$ | $10.13 \pm 2.11$ |
| FairAug | $67.04 \pm 0.69$ | $3.29 \pm 1.66$ | $\mathbf{3.04} \pm 1.37$ | $67.88 \pm 0.45$ | $4.81 \pm 0.59$ | $6.52 \pm 0.99$ |
| FairAug wo FM | $66.88 \pm 0.42$ | $4.64 \pm 1.91$ | $5.50 \pm 1.80$ | $66.15 \pm 0.75$ | $7.61 \pm 0.29$ | $9.28 \pm 1.06$ |
| FairAug wo NS | $67.01 \pm 0.41$ | $6.05 \pm 2.63$ | $6.83 \pm 2.88$ | $66.33 \pm 0.20$ | $6.26 \pm 0.26$ | $9.16 \pm 0.63$ |
| FairAug wo ED | $\mathbf{67.38} \pm 0.31$ | $\mathbf{2.66} \pm 3.22$ | $4.26 \pm 2.92$ | $67.57 \pm 0.18$ | $\mathbf{3.37} \pm 0.62$ | $\mathbf{5.13} \pm 1.12$ |
| FairAug wo EA | $66.96 \pm 1.08$ | $2.86 \pm 1.79$ | $3.22 \pm 1.29$ | $\mathbf{67.96} \pm 0.19$ | $7.13 \pm 0.29$ | $9.40 \pm 0.81$ |

values. Second, we note that similar to our framework, GCA is built upon GRACE through adaptive augmentations as well. However, the adaptive augmentations utilized in GCA are not fairness-aware, and the results of Table 1 demonstrate that the effect of such augmentations on the fairness metrics is unpredictable. Third, the results indicate that all contrastive learning methods provide better fairness performance than random walk-based methods on evaluated datasets, including Fairwalk, which is a fairness-aware study. Since the sole information source of random walk-based studies is the graph structure, obtained results confirm that the graph topology indeed propagates bias, which is consistent with the motivation of our graph data augmentation design. Finally, the results of Table 1 demonstrate that the closest competitor scheme to FairAug is NIFTY (Agarwal et al., 2021) in terms of fairness measures. For $\Delta_{SP}$, FairAug outperforms NIFTY on both datasets, whereas in terms of $\Delta_{EO}$, FairAug and NIFTY outperform each other on Pokec-z and Pokec-n, respectively. Table 1 shows that the ED-removed version of FairAug, 'FairAug wo ED', outperforms NIFTY on Pokec-n as well, suggesting that at least one of the proposed strategies outperform all considered benchmark schemes in both $\Delta_{SP}$ and $\Delta_{EO}$. This fairness performance improvement achieved by ED removal motivated us to conduct an ablation study on the building blocks of FairAug, which we present in the sequel.

Table 2 lists the results of an ablation study for FairAug, where the last four rows demonstrate the effects of the removal of FM, NS, ED, and EA, respectively. Pokec-z and Pokec-n are highly unbalanced datasets ($|\mathcal{E}^\omega| \approx 20|\mathcal{E}^\chi|$) with a considerable sparsity in inter-edges (see Table 5 of Appendix A.5). For such networks, the exact probabilities presented in equation 7 cannot be utilized for edge deletion as it would result in the deletion of a significantly large portion of the edges, damaging the graph structure. We employ an upper limit on the deletion probabilities to avoid this (see Appendix A.7). However, with such a limit, the proposed ED framework alone cannot sufficiently balance $|\mathcal{E}^\omega|$ and $|\mathcal{E}^\chi|$. At this point, we note that since $|\mathcal{S}^\omega| > |\mathcal{S}^\chi|$, node sampling also provides a way of decreasing $|\mathcal{E}^\omega|$ by excluding nodes from the set $\mathcal{S}^\omega$ when generating the subgraph. As the graph was already sparse initially in inter-edges (i.e., small $|\mathcal{E}^\chi|$), employing both NS and ED causes a similar over-deletion of intra-edges and creates unstable results due to the significantly distorted graph structure. Overall, combining this phenomenon with the results of Table 2, in order to balance the input graphs on Pokec networks consistently (which have $|\mathcal{E}^\omega| >> |\mathcal{E}^\chi|$ and inter-edge sparsity), node sampling is observed to be a better choice than edge manipulations.

Table 3 presents the obtained results for link prediction in the framework of contrastive learning. Results demonstrate that 'FairAug wo NS' consistently improves the fairness metrics of the framework it is built upon (GRACE), together with similar utility measures. In addition, comparing GCA

Table 3: Link prediction results obtained on node representations

| | UCSD34 | | | Berkeley13 | | |
|---|---|---|---|---|---|---|
| | AUC (%) | $\Delta_{SP}$ (%) | $\Delta_{EO}$ (%) | AUC (%) | $\Delta_{SP}$ (%) | $\Delta_{EO}$ (%) |
| GCA | $71.59 \pm 0.29$ | $0.70 \pm 0.27$ | $1.92 \pm 0.45$ | $69.69 \pm 0.38$ | $\mathbf{0.50} \pm 0.36$ | $6.52 \pm 0.99$ |
| GRACE | $71.57 \pm 0.28$ | $0.49 \pm 0.28$ | $1.48 \pm 0.70$ | $69.55 \pm 0.46$ | $0.76 \pm 0.44$ | $4.34 \pm 1.20$ |
| FairAug | $71.46 \pm 0.30$ | $0.71 \pm 0.38$ | $1.62 \pm 0.62$ | $69.48 \pm 0.29$ | $0.70 \pm 0.43$ | $4.24 \pm 0.90$ |
| FairAug wo NS | $71.50 \pm 0.31$ | $\mathbf{0.41} \pm 0.32$ | $\mathbf{1.16} \pm 0.65$ | $69.65 \pm 0.33$ | $0.68 \pm 0.34$ | $\mathbf{4.22} \pm 0.97$ |

Table 4: Employment of fair edge deletion as an edge dropout method

| | | Accuracy (%) | AUC (%) | $\Delta_{SP}$ (%) | $\Delta_{EO}$ (%) |
|---|---|---|---|---|---|
| **Cora** | Edge Dropout | $82.79 \pm 0.83$ | $90.52 \pm 0.52$ | $57.22 \pm 2.19$ | $36.18 \pm 4.53$ |
| | Fair ED | $80.38 \pm 1.14$ | $87.95 \pm 1.22$ | $48.78 \pm 2.58$ | $27.79 \pm 3.89$ |
| **Citeseer** | Edge Dropout | $78.30 \pm 1.17$ | $87.93 \pm 1.05$ | $43.05 \pm 2.39$ | $25.45 \pm 3.69$ |
| | Fair ED | $77.26 \pm 1.80$ | $86.79 \pm 1.39$ | $39.90 \pm 3.57$ | $25.02 \pm 6.93$ |
| **Pubmed** | Edge Dropout | $88.63 \pm 0.34$ | $95.14 \pm 0.16$ | $45.59 \pm 0.78$ | $15.81 \pm 0.92$ |
| | Fair ED | $87.48 \pm 0.54$ | $94.15 \pm 0.36$ | $40.90 \pm 1.10$ | $11.70 \pm 0.68$ |

and GRACE, obtained results confirm our previous assessment regarding the unpredictable effect of GCA's augmentations on the fairness metrics. Finally, comparing FairAug with and without NS, the results of Table 3 show that in UCSD34 and Berkeley13, the employment of node sampling can be ineffective in improving fairness metrics, or can even worsen them. In UCSD34 and Berkeley13, we have $|\mathcal{S}^{\chi}| >> |\mathcal{S}^{\omega}|$ (see Table 5 of Appendix A.5). Therefore, for NS on these datasets, half of the nodes are sampled randomly from the sets $\mathcal{S}_1^{\chi}$ and $\mathcal{S}_0^{\chi}$ (as the limit on the minimum node sampling budget is half of the initial group size to avoid a possible over-sampling, see Appendix A.7). Since $|\mathcal{S}^{\chi}| >> |\mathcal{S}^{\omega}|$, such a sampling framework coincides with random sampling, which makes the effects of the proposed node sampling scheme unpredictable on the fairness metrics. Furthermore, while $\gamma_1$ suggests that the cardinality of the set $|\mathcal{S}^{\chi}|$ should be reduced when $|\mathcal{S}^{\omega}| \leq |\mathcal{S}^{\chi}|$, $\gamma_1$ actually appears in the upper bound and not in the exact $\|\boldsymbol{\rho}\|_1$ expression. The removal of nodes with inter-edges is actually counter-intuitive, as inter-edges generally help to reduce bias in graphs where $|\mathcal{E}^{\omega}| \geq |\mathcal{E}^{\chi}|$ (which holds for all datasets considered herein). Therefore, the proposed node sampling becomes effective for graph structures providing $|\mathcal{S}^{\omega}| \geq |\mathcal{S}^{\chi}|$. The effect of node sampling on Facebook networks is further investigated through $\gamma_1$ and $\gamma_2$ in Appendix A.8, which corroborates the explanations on the random/detrimental effect of node sampling for these datasets.

As also noted in Section 1, even though FairAug is presented through its application on graph constrastive learning in this section, the proposed approach can be fully or partially used in conjunction to other learning frameworks as well. To exemplify such a use case, Table 4 lists the results for an end-to-end link prediction task with a two-layer GCN where the proposed fair edge deletion scheme is employed as an edge dropout method. As the benchmark, random edge dropout (Rong et al., 2019) is considered, which is a scheme originally proposed to improve the generalizability of the model on unseen data (Rong et al., 2019). In the experiments, the number of deleted edges is the same for both strategies. The results demonstrate that our method 'Fair ED' can indeed enhance the fairness of the learning framework it is employed in while it slightly reduces the utility measures.

## 5 CONCLUSIONS

In this study, the source of bias in aggregated representations in a GNN-based framework has been theoretically analyzed. Based on the analysis, several fairness-aware augmentation schemes have been introduced on both graph structure and nodal features. The proposed augmentations can be flexibly utilized together with several GNN-based learning methods. In addition, they can readily be employed in unsupervised node representation learning schemes such as graph contrastive learning. Experimental results on real-world graphs demonstrate that the proposed adaptive augmentations can improve fairness metrics with comparable utilities to state-of-the-art in node classification and link prediction.

## 6 REPRODUCIBILITY STATEMENT

The proofs for Theorem 1 and Proposition 1 are presented together with all assumptions made for them in Appendix A.1 and Appendix A.2, respectively. Implementation details and hyperparameter values are presented in Appendix A.7 for the clarification of experimental settings. All codes that are needed to regenerate the presented results in Tables 1, 2, 3 and 4 are provided within the supplementary material together with a README file. Finally, the details on the utilized data sets are provided in Appendix A.5 where all datasets are publicly available.

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

# A APPENDIX

## A.1 PROOF OF THEOREM 1

Let $\bar{s} = \frac{1}{N}\sum_{i=1}^{N} s_i = \frac{|\mathcal{S}_1|}{N}$ Hence, the elements of centered sensitive attribute vector can be written as

$$\rho_i = \frac{1}{N\sigma_\mathbf{s}\sigma_{\tilde{\mathbf{z}}_{:,i}}} \left( \sum_{v_m \in S_0} (0 - \bar{s})(z_{m,i} - \bar{z}_i) + \sum_{v_n \in S_1} (1 - \bar{s})(z_{n,i} - \bar{z}_i) \right), \tag{8}$$

where $\bar{z}_i = \frac{1}{N}\sum_{j=1}^{N} z_{ji}$ and $z_{ji}$ is the element of matrix $\mathbf{Z}$ at row $j$, column $i$. Using $\bar{s}$ values, the equation 8 becomes

$$
\begin{aligned}
\rho_i &= \frac{1}{N\sigma_\mathbf{s}\sigma_{\tilde{\mathbf{z}}_{:,i}}} \left( \sum_{v_m \in S_0} \frac{-|\mathcal{S}_1|}{N}(z_{m,i} - \bar{z}_i) + \sum_{v_n \in S_1} \frac{|\mathcal{S}_0|}{N}(z_{n,i} - \bar{z}_i) \right), \\
&= \frac{1}{N\sigma_\mathbf{s}\sigma_{\tilde{\mathbf{z}}_{:,i}}} \left( \frac{|\mathcal{S}_0||\mathcal{S}_1|}{N}\bar{z}_i + \sum_{v_m \in S_0} \frac{-|\mathcal{S}_1|}{N}z_{m,i} - \frac{|\mathcal{S}_0||\mathcal{S}_1|}{N}\bar{z}_i + \sum_{v_n \in S_1} \frac{|\mathcal{S}_0|}{N}z_{n,i} \right), \\
&= \frac{1}{N\sigma_\mathbf{s}\sigma_{\tilde{\mathbf{z}}_{:,i}}} \left( \sum_{v_n \in S_1} \frac{|\mathcal{S}_0|}{N}z_{n,i} - \sum_{v_m \in S_0} \frac{|\mathcal{S}_1|}{N}z_{m,i} \right), \\
&= \frac{|\mathcal{S}_0||\mathcal{S}_1|}{N^2\sigma_\mathbf{s}\sigma_{\tilde{\mathbf{z}}_{:,i}}} \left( \frac{1}{|\mathcal{S}_1|} \sum_{v_n \in S_1} z_{n,i} - \frac{1}{|\mathcal{S}_0|} \sum_{v_m \in S_0} z_{m,i} \right).
\end{aligned} \tag{9}
$$

Similarly analysis for $\sigma_\mathbf{s}$ follows as

$$
\begin{aligned}
\sigma_\mathbf{s} &:= \sqrt{\frac{1}{N}\sum_{n=1}^{N}(s_n - \bar{s})^2} = \sqrt{\frac{1}{N}\sum_{n=1}^{N}\left( \sum_{v_m \in S_0} \frac{|\mathcal{S}_1|^2}{N^2} + \sum_{v_n \in S_1} \frac{|\mathcal{S}_0|^2}{N^2} \right)} \\
&= \sqrt{\frac{1}{N}\left( \frac{|\mathcal{S}_0||\mathcal{S}_1|^2}{N^2} + \frac{|\mathcal{S}_1||\mathcal{S}_0|^2}{N^2} \right)} \\
&= \sqrt{\frac{1}{N}\left( |\mathcal{S}_0||\mathcal{S}_1|\frac{(|\mathcal{S}_0| + |\mathcal{S}_1|)}{(|\mathcal{S}_0| + |\mathcal{S}_1|)^2} \right)} \\
&= \frac{\sqrt{|\mathcal{S}_0||\mathcal{S}_1|}}{N}
\end{aligned} \tag{10}
$$

Define $c_i := \frac{\sqrt{|\mathcal{S}_0||\mathcal{S}_1|}}{N\sigma_{\tilde{\mathbf{z}}_{:,i}}}$ and $\mathbf{c} \in \mathbb{R}^{F'}$ denote the vector whose elements are $c_i$s. Then $\boldsymbol{\rho}$ equals to

$$\boldsymbol{\rho} = \mathbf{c} \circ \left( \frac{1}{|\mathcal{S}_1|} \sum_{v_n \in S_1} \mathbf{z}_n - \frac{1}{|\mathcal{S}_0|} \sum_{v_m \in S_0} \mathbf{z}_m \right) \tag{11}$$

where $\circ$ represents the Hadamard product. Therefore, $\|\boldsymbol{\rho}\|_1$ follows as

$$\|\boldsymbol{\rho}\|_1 = \left\| \mathbf{c} \circ \left( \frac{1}{|\mathcal{S}_1|} \sum_{v_n \in S_1} \mathbf{z}_n - \frac{1}{|\mathcal{S}_0|} \sum_{v_m \in S_0} \mathbf{z}_m \right) \right\|_1. \tag{12}$$

We first consider the terms $\frac{1}{|\mathcal{S}_1|}\sum_{v_n \in S_1} \mathbf{z}_n$ and $\frac{1}{|\mathcal{S}_0|}\sum_{v_m \in S_0} \mathbf{z}_m$ individually

$$
\begin{aligned}
\frac{1}{|\mathcal{S}_1|}\sum_{v_n \in S_1} \mathbf{z}_n &\in \frac{1}{|\mathcal{S}_1|}\sum_{v_n \in S_1}\left( \frac{1}{d_n}\sum_{a \in \mathcal{N}(n) \cap S_0} \boldsymbol{\mu}_0 + \frac{1}{d_n}\sum_{b \in \mathcal{N}(n) \cap S_1} \boldsymbol{\mu}_1 \right) \pm \Delta, \\
&\in \frac{1}{|\mathcal{S}_1|}\sum_{v_n \in S_1} \frac{d_n^\chi \boldsymbol{\mu}_0 + d_n^\omega \boldsymbol{\mu}_1}{d_n^\chi + d_n^\omega} \pm \Delta, \\
&\in \frac{1}{|\mathcal{S}_1|}\sum_{v_n \in S_1}\left( \boldsymbol{\mu}_1 + \frac{d_n^\chi}{d_n^\chi + d_n^\omega}(\boldsymbol{\mu}_0 - \boldsymbol{\mu}_1) \right) \pm \Delta.
\end{aligned} \tag{13}
$$

Similarly, the expression for the term $\frac{1}{|\mathcal{S}_0|}\sum_{v_m \in S_0} \mathbf{z}_m$ can also be derived.

$$\frac{1}{|\mathcal{S}_0|}\sum_{v_m \in S_0} \mathbf{z}_m \in \frac{1}{|\mathcal{S}_0|}\sum_{v_m \in S_0} \left(\boldsymbol{\mu}_0 + \frac{d_m^{\chi}}{d_m^{\chi} + d_m^{\omega}}(\boldsymbol{\mu}_1 - \boldsymbol{\mu}_0)\right) \pm \Delta. \tag{14}$$

Define $\boldsymbol{\epsilon} := \frac{1}{|\mathcal{S}_1|}\sum_{v_n \in S_1} \mathbf{z}_n - \frac{1}{|\mathcal{S}_0|}\sum_{v_m \in S_0} \mathbf{z}_m$, the following can be written by equation 13 and equation 14

$$\boldsymbol{\epsilon} \in (\boldsymbol{\mu}_1 - \boldsymbol{\mu}_0)\left(1 - \frac{1}{|\mathcal{S}_1|}\sum_{v_n \in S_1}\left(\frac{d_n^{\chi}}{d_n^{\chi} + d_n^{\omega}}\right) - \frac{1}{|\mathcal{S}_0|}\sum_{v_m \in S_0}\left(\frac{d_m^{\chi}}{d_m^{\chi} + d_m^{\omega}}\right)\right) \pm 2\Delta. \tag{15}$$

$\boldsymbol{\epsilon}$ is bounded below, which follows as

$$1 - \frac{1}{|\mathcal{S}_1|}\sum_{v_n \in S_1}\left(\frac{d_n^{\chi}}{d_n^{\chi} + d_n^{\omega}}\right) - \frac{1}{|\mathcal{S}_0|}\sum_{v_m \in S_0}\left(\frac{d_m^{\chi}}{d_m^{\chi} + d_m^{\omega}}\right)$$

$$= 1 - \frac{1}{|\mathcal{S}_1|}\sum_{v_n \in S_1^{\chi}}\left(\frac{d_n^{\chi}}{d_n^{\chi} + d_n^{\omega}}\right) - \frac{1}{|\mathcal{S}_0|}\sum_{v_m \in S_0^{\chi}}\left(\frac{d_m^{\chi}}{d_m^{\chi} + d_m^{\omega}}\right) \tag{16}$$

$$\geq 1 - \frac{|\mathcal{S}_0^{\chi}|}{|\mathcal{S}_0|} - \frac{|\mathcal{S}_1^{\chi}|}{|S_1|} \tag{17}$$

equation 16 follows as $d_u^{\chi} = 0$ for $\forall v_u \in \mathcal{S}^{\omega}$. The upper bound equation 17 holds, since $\left(\frac{d_u^{\chi}}{d_u^{\chi} + d_u^{\omega}}\right) \leq 1$ for $\forall v_u \in \mathcal{G}$.

The upper bound for $\boldsymbol{\epsilon}$ follows as

$$1 - \frac{1}{|\mathcal{S}_1|}\sum_{v_n \in S_1}\left(\frac{d_n^{\chi}}{d_n^{\chi} + d_n^{\omega}}\right) - \frac{1}{|\mathcal{S}_0|}\sum_{v_m \in S_0}\left(\frac{d_m^{\chi}}{d_m^{\chi} + d_m^{\omega}}\right) \tag{18}$$

$$= 1 - \text{mean}\left(\frac{d_m^{\chi}}{d_m^{\chi} + d_m^{\omega}}|v_m \in \mathcal{S}_0\right) - \text{mean}\left(\frac{d_n^{\chi}}{d_n^{\chi} + d_n^{\omega}}|v_n \in \mathcal{S}_1\right) \tag{19}$$

$$\leq 1 - 2\min\left(\text{mean}\left(\frac{d_m^{\chi}}{d_m^{\chi} + d_m^{\omega}}|v_m \in \mathcal{S}_0\right), \text{mean}\left(\frac{d_n^{\chi}}{d_n^{\chi} + d_n^{\omega}}|v_n \in \mathcal{S}_1\right)\right), \tag{20}$$

where $\text{mean}(\cdot)$ denotes the sample mean operation. The main purpose is to generate an upper bound for $\|\boldsymbol{\rho}\|_1$, for which the first step is to derive the following bound

$$\|\boldsymbol{\rho}\|_1 \leq \|\mathbf{c}\|_1\|\boldsymbol{\epsilon}\|_1 \tag{21}$$

equation 21 can be written, as $\|\mathbf{x} \circ \mathbf{y}\|_1 \leq \|\mathbf{x}\|_1\|\mathbf{y}\|_1$ for arbitrary vectors $\mathbf{x}$ and $\mathbf{y} \in \mathbb{R}^N$ ($|x_1 y_1| + \cdots + |x_N y_N| \leq (|x_1| + \cdots + |x_N|)(|y_1| + \cdots + |y_N|)$) due to additional non-negative cross-terms $|x_i||y_j|, i \neq j$). The next step is to bound the $\ell_1$ norm of the difference term $\boldsymbol{\epsilon}$:

$$\|\boldsymbol{\epsilon}\|_1 \leq \|\boldsymbol{\mu}_1 - \boldsymbol{\mu}_0\|_1\max(\gamma_1, \gamma_2) + 2N\Delta \tag{22}$$

using the derived bounds in equation 17, equation 20 and the equality $2\Delta \cdot \|\mathbf{1}\|_1 = 2N\Delta$, where $\mathbf{1}$ denotes an $N \times 1$ vector of ones, $\gamma_1 := \left|1 - \frac{|\mathcal{S}_0^{\chi}|}{|\mathcal{S}_0|} - \frac{|\mathcal{S}_1^{\chi}|}{|S_1|}\right|$, and $\gamma_2 = \left|1 - 2\min\left(\text{mean}\left(\frac{d_m^{\chi}}{d_m^{\chi} + d_m^{\omega}}|v_m \in \mathcal{S}_0\right), \text{mean}\left(\frac{d_n^{\chi}}{d_n^{\chi} + d_n^{\omega}}|v_n \in \mathcal{S}_1\right)\right)\right|$. Using equation 21 and equation 22, the final upper bound can be derived:

$$\|\boldsymbol{\rho}\|_1 \leq \|\mathbf{c}\|_1\left(\|\boldsymbol{\mu}_1 - \boldsymbol{\mu}_0\|_1\max(\gamma_1, \gamma_2) + 2N\Delta\right). \tag{23}$$

## A.2 PROOF OF PROPOSITION 1

The expected $\|\tilde{\boldsymbol{\delta}}\|_1$ can be written as:

$$
E_{\mathbf{p}}[\|\tilde{\boldsymbol{\delta}}\|_1] = E\left[\sum_{i=1}^{F}|\tilde{\delta}_i|\right]
$$
$$
= \sum_{i=1}^{F} E[|\tilde{\delta}_i|] = \sum_{i=1}^{F} p_i|\delta_i|. \tag{24}
$$

Similarly, the expected $\|\tilde{\boldsymbol{\delta}}\|_1$ for the uniform masking scheme is:[1]

$$
E_{\mathbf{q}}[\|\tilde{\boldsymbol{\delta}}\|_1] = \sum_{i=1}^{F} q_i|\delta_i|. \tag{25}
$$

For fair comparison, set uniform keeping probability of the nodal features $q_i = \bar{p} = \frac{1}{F}\sum_{i=1}^{F} p_i$. Without loss of generality, assume, $|\delta_i|$s are ordered such that $|\delta_1| \leq \cdots \leq |\delta_F|$. With the ordered $|\delta_i|$s, assigned probabilities to keep the nodal features by our method will also be ordered such that $p_1 \geq \cdots \geq p_F$. Defining a dummy variable $|\delta_0| := 0$, equation 24 can be rewritten as:

$$
\sum_{i=1}^{F} p_i|\delta_i| = (|\delta_1| - |\delta_0|)(p_1 + \cdots + p_F) + (|\delta_2| - |\delta_1|)(p_2 + \cdots + p_F) + \ldots
$$
$$
\cdots + (|\delta_{F-1}| - |\delta_{F-2}|)(p_{F-1} + p_F) + (|\delta_F| - |\delta_{F-1}|)(p_F) \tag{26}
$$
$$
= \sum_{l=1}^{F}(|\delta_l| - |\delta_{l-1}|)\sum_{i=l}^{F} p_i.
$$

Following the definition in (Marshall et al., 1979), a sequence $\boldsymbol{x} = \{x_1, x_2, \ldots, x_F\}$ majorizes another sequence $\boldsymbol{y} = \{y_1, y_2, \ldots, y_F\}$, if the following holds:

$$
\sum_{i=1}^{k} y_i \leq \sum_{i=1}^{k} x_i \quad k = 1, \ldots, F-1,
$$
$$
\sum_{i=1}^{F} y_i = \sum_{i=1}^{F} x_i. \tag{27}
$$

Since the uniform sequence is majorized by any other non-increasing ordered sequence with the same sum (see (Gomez et al., 2019, Equation 3)), the sequence $\mathbf{p} = \{p_1, p_2, \cdots, p_F\}$ majorizes the sequence $\mathbf{q} = \{q_1, q_2, \ldots, q_F\}$ where $q_i = \bar{p}, \forall i \in \{1, \ldots, F\}$. Defining $\sum_{i=1}^{0} p_i := 0$, equation 26 can be re-written as

$$
E_{\mathbf{p}}[\|\tilde{\boldsymbol{\delta}}\|_1] = \sum_{l=1}^{F}(|\delta_l| - |\delta_{l-1}|)\sum_{i=l}^{F} p_i
$$
$$
= \sum_{l=1}^{F}(|\delta_l| - |\delta_{l-1}|)(F\bar{p} - \sum_{i=1}^{l-1} p_i)
$$
$$
\overset{(a)}{\leq} \sum_{l=1}^{F}(|\delta_l| - |\delta_{l-1}|)(F\bar{p} - \sum_{i=1}^{l-1} q_i) \tag{28}
$$
$$
= \sum_{l=1}^{F}(|\delta_l| - |\delta_{l-1}|)\sum_{i=l}^{F} q_i
$$
$$
= E_{\mathbf{q}}[\|\tilde{\boldsymbol{\delta}}\|_1],
$$

---

[1]From equation 25, it can be obtained that $E_{\mathbf{q}}[\|\tilde{\boldsymbol{\delta}}\|_1] = \sum_{i=1}^{F} q_i|\delta_i| \leq \sum_{i=1}^{F}|\delta_i| = \|\boldsymbol{\delta}\|_1$, as the probabilities $q_i \in [0, 1]$. Here, $\|\boldsymbol{\delta}\|_1$ corresponds to the distribution difference with no feature masking ($q_i = 1, \forall i$). This implies that no feature masking upper bounds $E[\|\tilde{\boldsymbol{\delta}}\|_1]$, which is obtained via a stochastic feature masking.

where inequality (a) follows from the definition of majorization in equation 27.

## A.3 NODE SAMPLING AND ITS EFFECTS

---

**Algorithm 1:** Node Sampling

---

**Data:** $\mathcal{G} := (\mathcal{V}, \mathcal{E}), \mathbf{X}, \mathbf{s}, \beta$
**Result:** $\tilde{\mathcal{G}}, \tilde{\mathbf{X}}, \tilde{\mathbf{s}}$
Split $\mathcal{V}$ into sets $\{\mathcal{S}_0^\chi, \mathcal{S}_1^\chi, \mathcal{S}_0^\omega, \mathcal{S}_1^\omega\}$
**if** $|\mathcal{S}^\omega| \geq |\mathcal{S}^\chi|$ **then**
     Sample $|\mathcal{S}_0^\chi|$ nodes from $\mathcal{S}_0^\omega$ uniformly to obtain $\bar{\mathcal{S}}_0^\omega$;
     Sample $|\mathcal{S}_1^\chi|$ nodes from $\mathcal{S}_1^\omega$ uniformly to obtain $\bar{\mathcal{S}}_1^\omega$;
     $\tilde{\mathcal{V}} := \{\mathcal{S}^\chi, \bar{\mathcal{S}}_0^\omega, \bar{\mathcal{S}}_1^\omega\}$;
**end**
**if** $|\mathcal{S}^\chi| \geq |\mathcal{S}^\omega|$ **then**
     Sample $|\mathcal{S}_0^\omega|$ nodes from $\mathcal{S}_0^\chi$ uniformly to obtain $\bar{\mathcal{S}}_0^\chi$;
     Sample $|\mathcal{S}_1^\omega|$ nodes from $\mathcal{S}_1^\chi$ uniformly to obtain $\bar{\mathcal{S}}_1^\chi$;
     $\tilde{\mathcal{V}} := \{\bar{\mathcal{S}}_0^\chi, \bar{\mathcal{S}}_1^\chi, \mathcal{S}^\omega\}$;
**end**
Obtain subgraph $\tilde{\mathcal{G}}$ induced by $\tilde{\mathcal{V}}$ along with the nodal features $\tilde{\mathbf{X}}$ and sensitive attributes $\tilde{\mathbf{s}}$

---

The overall node sampling scheme is presented in Algorithm 1.

Assuming $|\mathcal{S}^\omega| \geq |\mathcal{S}^\chi|$, all nodes in $\mathcal{S}^\chi$ are retained, meaning $\tilde{\mathcal{S}}_0^\chi = \mathcal{S}_0^\chi$ and $\tilde{\mathcal{S}}_1^\chi = \mathcal{S}_1^\chi$. While subsets of nodes $\bar{\mathcal{S}}_0^\omega$ and $\bar{\mathcal{S}}_1^\omega$ are randomly sampled from $\mathcal{S}_0^\omega$ and $\mathcal{S}_1^\omega$ respectively with sample size $|\bar{\mathcal{S}}_0^\omega| = |\mathcal{S}_0^\chi|$ and $|\bar{\mathcal{S}}_1^\omega| = |\mathcal{S}_1^\chi|$.

Therefore, in the resulting graph $\tilde{\mathcal{G}}$, we have

$$|\tilde{\mathcal{S}}_0^\omega| = |\bar{\mathcal{S}}_0^\omega| = |\mathcal{S}_0^\chi| \tag{29}$$

$$|\tilde{\mathcal{S}}_1^\omega| = |\bar{\mathcal{S}}_1^\omega| = |\mathcal{S}_1^\chi| \tag{30}$$

therefore

$$\frac{\left|\tilde{\mathcal{S}}_0^\chi\right|}{|\tilde{\mathcal{S}}_0^\chi| + |\tilde{\mathcal{S}}_0^\omega|} = \frac{|\mathcal{S}_0^\chi|}{|\mathcal{S}_0^\chi| + |\mathcal{S}_0^\chi|} = \frac{1}{2} \tag{31}$$

$$\frac{\left|\tilde{\mathcal{S}}_1^\chi\right|}{|\tilde{\mathcal{S}}_1^\chi| + |\tilde{\mathcal{S}}_1^\omega|} = \frac{|\mathcal{S}_1^\chi|}{|\mathcal{S}_1^\chi| + |\mathcal{S}_1^\chi|} = \frac{1}{2} \tag{32}$$

which leads to $\gamma_1 = 0$ for the obtained $\tilde{\mathcal{G}}$.

## A.4 EFFECTS OF GLOBAL EDGE DELETION SCHEME IN EQUATION 7

Given the edge deletion scheme in equation 7, we have

$$\mathbb{E}_{\mathbf{p}^{(e)}}\left[|\tilde{\mathcal{E}}^\chi|\right] = \pi|\mathcal{E}^\chi| \tag{33}$$

$$\mathbb{E}_{\mathbf{p}^{(e)}}\left[|\tilde{\mathcal{E}}_{\mathcal{S}_0}^\omega|\right] = \frac{|\mathcal{E}^\chi|}{2\left|\mathcal{E}_{\mathcal{S}_0}^\omega\right|}\pi \times \left|\mathcal{E}_{\mathcal{S}_0}^\omega\right| = \frac{\pi|\mathcal{E}^\chi|}{2} \tag{34}$$

$$\mathbb{E}_{\mathbf{p}^{(e)}}\left[|\tilde{\mathcal{E}}_{\mathcal{S}_1}^\omega|\right] = \frac{|\mathcal{E}^\chi|}{2\left|\mathcal{E}_{\mathcal{S}_1}^\omega\right|}\pi \times \left|\mathcal{E}_{\mathcal{S}_1}^\omega\right| = \frac{\pi|\mathcal{E}^\chi|}{2}. \tag{35}$$

Therefore, $\mathbb{E}_{\mathbf{p}^{(e)}}\left[|\tilde{\mathcal{E}}_{\mathcal{S}_0}^\omega|\right] = \mathbb{E}_{\mathbf{p}^{(e)}}\left[|\tilde{\mathcal{E}}_{\mathcal{S}_1}^\omega|\right] = \frac{1}{2}\mathbb{E}_{\mathbf{p}^{(e)}}\left[|\tilde{\mathcal{E}}^\chi|\right]$ and equation 6 holds.

## A.5 DATASET STATISTICS

Dataset statistics are provided in Tables 5 and 6.

Table 5: Dataset statistics for social networks

| Dataset | $|\mathcal{S}_0^X|$ | $|\mathcal{S}_0^\omega|$ | $|\mathcal{S}_1^X|$ | $|\mathcal{S}_1^\omega|$ | $|\mathcal{E}^X|$ | $|\mathcal{E}_{\mathcal{S}_0}^\omega|$ | $|\mathcal{E}_{\mathcal{S}_1}^\omega|$ |
|---------|------|------|------|------|------|------|------|
| Pokec-z | 622 | 4229 | 582 | 2226 | 1730 | 23428 | 15942 |
| Pokec-n | 423 | 3617 | 479 | 1666 | 1422 | 18548 | 10672 |
| UCSD34 | 2246 | 118 | 1697 | 71 | 51607 | 36787 | 19989 |
| Berkeley13 | 1619 | 80 | 1488 | 77 | 27542 | 19550 | 13582 |

Table 6: Dataset statistics for citation networks

| Dataset | $|\mathcal{V}|$ | # sensitive attr. | $|\mathcal{E}^X|$ | $|\mathcal{E}^\omega|$ |
|---------|------|------|------|------|
| Cora | 2708 | 7 | 1428 | 5964 |
| Citeseer | 3327 | 6 | 1628 | 4746 |
| Pubmed | 19717 | 3 | 12254 | 49802 |

## A.6 CONTRASTIVE LEARNING OVER GRAPHS

The main goal of contrastive learning is to learn discriminable representations by contrasting the embeddings of positive and negative examples, through minimizing a specific contrastive loss (Opolka et al., 2019; Veličković et al., 2019; Zhu et al., 2021; 2020). The contrastive loss in the present work is designed to maximize node-level agreement, meaning that the representations of the same node generated from different graph views can be discriminated from the embeddings of other nodes. Let $\mathbf{H}^1 = f(\widetilde{\mathbf{A}}^1, \widetilde{\mathbf{X}}^1)$ and $\mathbf{H}^2 = f(\widetilde{\mathbf{A}}^2, \widetilde{\mathbf{X}}^2)$ denote the nodal embeddings generated with graph views $\widetilde{G}^1$ and $\widetilde{G}^2$, where $\widetilde{\mathbf{A}}^i$, and $\widetilde{\mathbf{X}}^i$ are the adjacency and feature matrices of $\widetilde{G}^i$, which are corrupted versions of the matrices $\mathbf{A}$ and $\mathbf{X}$. Let $\mathbf{h}_i^1$ and $\mathbf{h}_i^2$ denote the embeddings for $v_i$: They should be more similar to each other than to the embeddings of all other nodes. Hence, the representations of all other nodes are used as negative samples. The contrastive loss for generating embeddings $\mathbf{h}_i^1$ and $\mathbf{h}_i^2$ (considering $\mathbf{h}_i^1$ as the anchor representation) can be written as

$$\ell\left(\mathbf{h}_i^1, \mathbf{h}_i^2\right) = -\log \frac{e^{s(\mathbf{h}_i^1, \mathbf{h}_i^2)/\tau}}{e^{s(\mathbf{h}_i^1, \mathbf{h}_i^2)/\tau} + \sum_{k=1}^N 1_{[k \neq i]} e^{s(\mathbf{h}_i^1, \mathbf{h}_k^2)/\tau} + \sum_{k=1}^N 1_{[k \neq i]} e^{s(\mathbf{h}_i^1, \mathbf{h}_k^1)/\tau}} \tag{36}$$

where $s\left(\mathbf{h}_i^1, \mathbf{h}_i^2\right) := c\left(g(\mathbf{h}_i^1), g(\mathbf{h}_i^2)\right)$, with $c(\cdot, \cdot)$ denoting the cosine similarity between the input vectors, and $g(\cdot)$ representing a nonlinear learnable transform executed by utilizing a 2-layer multi-layer perceptron (MLP), see also, (Zhu et al., 2021; 2020; Chen et al., 2020b). $\tau$ denotes the temperature parameter, and $1_{[k \neq i]} \in \{0, 1\}$ is the indicator function which takes the value 1 when $k \neq i$. This objective can also be written in the symmetric form, when the generated representation $\mathbf{h}_i^2$ is considered as the anchor example. Therefore, considering all nodes in the given graph, the final loss can be expressed as

$$\mathcal{J} = \frac{1}{2N} \sum_{i=1}^N \left[\ell\left(\mathbf{h}_i^1, \mathbf{h}_i^2\right) + \ell\left(\mathbf{h}_i^2, \mathbf{h}_i^1\right)\right]. \tag{37}$$

## A.7 HYPERPARAMETERS AND IMPLEMENTATION DETAILS

**Contrastive Learning:** All experimental settings are kept unaltered as presented in our baseline study GRACE (Zhu et al., 2020) for fair comparison, where in the GNN-based encoder, weights are initialized utilizing Glorot initialization (Glorot & Bengio, 2010) and ReLU activation is used after each GCN layer. All models are trained for 400 epochs by employing Adam optimizer (Kingma & Ba, 2014) together with a learning rate of $5 \times 10^{-4}$ and $\ell_2$ weight decay factor of $10^{-5}$. The temperature parameter $\tau$ in contrastive loss is chosen to be $0.4$ and the dimension of the node representations is selected as 256 for all datasets.

**End-to-end training:** All models are trained for $100$ epochs by employing Adam optimizer (Kingma & Ba, 2014) together with a learning rate of $5 \times 10^{-3}$. The dimension of the node representations is selected as $128$ for all datasets.

**Implementation details:** Node representations are obtained using a two-layer GCN (Kipf & Welling, 2017) for all contrastive learning baselines, which is kept the same as the one used in (Zhu et al., 2021; 2020) to ensure fair comparison. After obtaining node embeddings from different schemes, a linear classifier based on $\ell_2$-regularized logistic regression is applied for the node classification and link prediction tasks, which is again the same as the scheme used in (Veličković et al., 2019; Zhu et al., 2021; 2020). In node classification, the classifier is trained on $90\%$ of the nodes selected through a random split, and the remaining nodes constitute the test set. For each experiment, results for three random data splits are obtained, and the average together with standard deviations are presented. In link prediction, $90\%$ of the edges are utilized in the training of GCN encoder and linear classifier. For each edge, the input to the linear classifier is the concatenated representations of the nodes that the edge connects. Results for link prediction are obtained for five different edge splits, the resulting average metrics and standard deviations are provided. Finally, in the end-to-end link prediction experiments, a two-layer GCN is trained, and results are obtained for 6 different random data splits.

Note that if the sizes of the sets $\hat{\mathcal{S}}$ and $\tilde{\mathcal{S}}$ are highly unbalanced, we put a limit on the minimum of sampling budgets in node sampling to prevent any excessive corruption of the input graph. If $\mathcal{S}^{\chi} \geq \mathcal{S}^{\omega}$, the minimum limit for each set is the $50\%$ of the initial group size, otherwise the limit is the $25\%$ of the initial group size. Furthermore, to avoid overly down sampling the edges, a minimum value for $p^{(e)}(e_{ij})$, $p^{(e)}_{min}$, is set for the edge deletion scheme. In the feature masking scheme, $\alpha$ in equation 2 is set to $0$ and $0.1$ for different graph views used in the contrastive loss (i.e., default parameter values provided for GCA are used directly). In the edge deletion framework, $\pi = 1$, and $p^{(e)}_{min} = \frac{\pi}{2} = 0.5$ are used in the experiments of both Pokec datasets and citation networks. Such a selection is utilized to present the scheme with the most "natural" hyperparameter value, in the sense that only intra-edges are deleted, where the already sparse inter-edges are left unchanged. We note that this selection is naturally a good choice for unbalanced data sets in terms of inter- and intra-edges. For Facebook networks where intra- and inter-edges are balanced for certain sensitive groups, $\pi = 0.8$, and $p^{(e)}_{min}$ is set as $\frac{\pi}{2}$ for all experiments.

Proposed augmentation frameworks have only two hyperparameters: $\alpha$ in feature masking and $\pi$ in adaptive edge deletion. While we note the value of $\alpha$ is kept the same for all baselines that utilize a feature masking scheme (default hyperparameters provided by GCA (Zhu et al., 2021)), and 1 is the most natural choice for $\pi$ in Pokec datasets and citation networks, we carry out a sensitivity analysis for these two hyperparameters in this appendix. In the analysis, for $\alpha$, the values $[0.1, 0.2, 0.3]$ are examined for the first graph view, where the $\alpha$ value for the second view is assigned to be higher than the value of the first one by $0.1$. Overall, sensitivity analyses for Pokec and Facebook networks on the FairAug algorithm are presented in Tables 7 and 8, respectively. In Tables 7 and 8, the first column shows the $\alpha$ values of the first and second view, respectively.

The sensitivity analysis for $\alpha$ on Pokec networks in Table 7 demonstrates that increased $\alpha$ values can both improve or degrade the fairness performance together with similar classification accuracies. However, the resulted fairness performances are observed to be consistently better than our baseline, GRACE. Furthermore, the results of Table 8 show that fairness performances for the link prediction task can generally be improved with increased $\alpha$ values, and the performance is quite stable over different $\alpha$ selections.

Table 7: Sensitivity analysis for $\alpha$ on Pokec networks for FairAug.

| $\alpha$ | Pokec-z | | | Pokec-n | | |
|---|---|---|---|---|---|---|
| | Accuracy (%) | $\Delta_{SP}$ (%) | $\Delta_{EO}$ (%) | Accuracy (%) | $\Delta_{SP}$ (%) | $\Delta_{EO}$ (%) |
| 0.0/0.1 | $67.04 \pm 0.69$ | $3.29 \pm 1.66$ | $3.04 \pm 1.37$ | $67.88 \pm 0.45$ | $4.81 \pm 0.59$ | $6.52 \pm 0.99$ |
| 0.1/0.2 | $67.90 \pm 0.69$ | $3.95 \pm 2.09$ | $4.20 \pm 2.41$ | $67.60 \pm 0.48$ | $4.38 \pm 0.45$ | $5.82 \pm 1.44$ |
| 0.2/0.3 | $67.68 \pm 0.74$ | $4.42 \pm 1.48$ | $4.46 \pm 2.25$ | $68.01 \pm 0.36$ | $4.95 \pm 0.39$ | $6.44 \pm 1.36$ |
| 0.3/0.4 | $67.71 \pm 0.64$ | $2.99 \pm 1.89$ | $3.20 \pm 2.57$ | $67.50 \pm 0.11$ | $3.38 \pm 0.90$ | $4.41 \pm 1.12$ |

Table 8: Sensitivity analysis for $\alpha$ on Facebook networks for FairAug

| $\alpha$ | UCSD34 | | | Berkeley13 | | |
|---|---|---|---|---|---|---|
| | AUC (%) | $\Delta_{SP}$ (%) | $\Delta_{EO}$ (%) | AUC (%) | $\Delta_{SP}$ (%) | $\Delta_{EO}$ (%) |
| 0.0/0.1 | $71.46 \pm 0.30$ | $0.71 \pm 0.38$ | $1.62 \pm 0.62$ | $69.48 \pm 0.29$ | $0.70 \pm 0.43$ | $4.24 \pm 0.90$ |
| 0.1/0.2 | $71.54 \pm 0.33$ | $0.65 \pm 0.13$ | $1.76 \pm 0.38$ | $69.50 \pm 0.42$ | $0.55 \pm 0.22$ | $4.31 \pm 0.76$ |
| 0.2/0.3 | $71.51 \pm 0.30$ | $0.58 \pm 0.26$ | $1.50 \pm 0.66$ | $69.57 \pm 0.46$ | $0.55 \pm 0.35$ | $4.15 \pm 0.79$ |
| 0.3/0.4 | $71.55 \pm 0.41$ | $0.49 \pm 0.14$ | $1.65 \pm 0.50$ | $69.41 \pm 0.35$ | $0.97 \pm 0.19$ | $4.47 \pm 1.10$ |

The sensitivity analysis for $\pi$ is carried out by examining the values $[0.75, 0.80, 0.85, 0.90, 0.95, 1.00]$ for FairAug algorithm for both views. The sensitivity analyses for Pokec and Facebook networks on FairAug algorithm are presented in Tables 9 and 10, respectively. In Tables 9 and 10, $\pi = 1$ is the parameter choice utilized to generate the results in Table 2 and $\pi = 0.8$ is the selection for the results in Table 3.

Table 9: Sensitivity analysis for $\pi$ on Pokec networks for FairAug.

| $\pi$ | Pokec-z | | | Pokec-n | | |
|---|---|---|---|---|---|---|
| | Accuracy (%) | $\Delta_{SP}$ (%) | $\Delta_{EO}$ (%) | Accuracy (%) | $\Delta_{SP}$ (%) | $\Delta_{EO}$ (%) |
| 1.00 | $67.04 \pm 0.69$ | $3.29 \pm 1.66$ | $3.04 \pm 1.37$ | $67.88 \pm 0.45$ | $4.81 \pm 0.59$ | $6.52 \pm 0.99$ |
| 0.95 | $67.66 \pm 0.57$ | $4.00 \pm 1.76$ | $4.37 \pm 1.82$ | $68.01 \pm 0.53$ | $5.66 \pm 1.27$ | $7.00 \pm 0.72$ |
| 0.90 | $67.14 \pm 0.73$ | $3.28 \pm 1.97$ | $3.22 \pm 2.20$ | $67.81 \pm 0.33$ | $5.22 \pm 0.82$ | $6.87 \pm 1.22$ |
| 0.85 | $66.96 \pm 0.77$ | $3.49 \pm 2.28$ | $3.52 \pm 2.15$ | $68.00 \pm 0.29$ | $5.66 \pm 0.29$ | $7.77 \pm 1.01$ |
| 0.80 | $67.87 \pm 0.35$ | $3.59 \pm 2.33$ | $4.30 \pm 1.95$ | $68.17 \pm 0.28$ | $5.61 \pm 0.73$ | $7.82 \pm 1.24$ |
| 0.75 | $67.74 \pm 0.28$ | $3.94 \pm 2.48$ | $4.58 \pm 1.91$ | $67.95 \pm 0.10$ | $5.69 \pm 0.27$ | $7.44 \pm 1.08$ |

Table 10: Sensitivity analysis for $\pi$ on Facebook networks for FairAug

| $\pi$ | UCSD34 | | | Berkeley13 | | |
|---|---|---|---|---|---|---|
| | AUC (%) | $\Delta_{SP}$ (%) | $\Delta_{EO}$ (%) | AUC (%) | $\Delta_{SP}$ (%) | $\Delta_{EO}$ (%) |
| 0.80 | $71.46 \pm 0.30$ | $0.71 \pm 0.38$ | $1.62 \pm 0.62$ | $69.48 \pm 0.29$ | $0.70 \pm 0.43$ | $4.24 \pm 0.90$ |
| 1.00 | $71.30 \pm 0.41$ | $0.56 \pm 0.47$ | $1.41 \pm 0.57$ | $69.70 \pm 0.32$ | $0.74 \pm 0.36$ | $4.35 \pm 1.11$ |
| 0.95 | $71.37 \pm 0.41$ | $0.60 \pm 0.30$ | $1.48 \pm 0.69$ | $69.52 \pm 0.42$ | $0.66 \pm 0.16$ | $3.99 \pm 0.84$ |
| 0.90 | $71.40 \pm 0.38$ | $0.59 \pm 0.34$ | $1.45 \pm 0.44$ | $69.48 \pm 0.31$ | $0.86 \pm 0.72$ | $4.64 \pm 1.37$ |
| 0.85 | $71.39 \pm 0.36$ | $0.65 \pm 0.28$ | $1.38 \pm 0.54$ | $69.49 \pm 0.41$ | $0.70 \pm 0.40$ | $4.19 \pm 0.84$ |
| 0.75 | $71.47 \pm 0.23$ | $0.63 \pm 0.37$ | $1.55 \pm 0.65$ | $69.52 \pm 0.43$ | $0.74 \pm 0.48$ | $4.31 \pm 1.19$ |

The sensitivity analysis for $\pi$ on Pokec networks shows that $\pi = 1$ results in the best performances in terms of fairness. However, we note that the fairness results for the remaining $\pi$ values also outperform our baseline, GRACE, together with similar node classification accuracies. Furthermore, the results of Table 10 demonstrate that the presented results for link prediction in Table 3 can indeed be improved with a grid search, as better fairness performances can be obtained with $\pi \neq 0.8$. Moreover, the results for different $\pi$ values vary less for link prediction compared to node classification.

A.8 Effects of Proposed Augmentations on $\gamma_1$ and $\gamma_2$

Table 11: Effects of proposed augmentations on $\gamma_2$

| | Original | Node Sampling | Edge Deletion | Edge Addition |
|---|---|---|---|---|
| Pokec-z | 0.90 | 0.59 | 0.87 | 0.77 |
| Pokec-n | 0.91 | 0.62 | 0.89 | 0.81 |
| UCSD34 | 0.18 | 0.45 | 0.03 | 0.11 |
| Berkeley13 | 0.18 | 0.43 | 0.03 | 0.06 |

Tables 11 and 12 present the effects of the proposed framework on $\gamma$ values for different datasets. In Table 11, the effect of each augmentation step on $\gamma_2$ is considered independently. Table 12 demonstrates the effect of augmentations sequentially in the overall algorithm in a cumulative manner (e.g., the "Edge Deletion" column implies both Node Sampling and Edge Deletion are applied). Both Tables 11 and 12 demonstrate that even though the edge deletion/addition schemes are not

Table 12: Effects of each step in FairAug

| | | Original Graph | Node Sampling | Edge Deletion | Edge Addition |
|---|---|---|---|---|---|
| Pokec-z | $\gamma_1$ | 0.66 | 0.11 | 0.11 | 0.11 |
| | $\gamma_2$ | 0.90 | 0.59 | 0.50 | 0.43 |
| Pokec-n | $\gamma_1$ | 0.67 | 0.15 | 0.15 | 0.15 |
| | $\gamma_2$ | 0.91 | 0.62 | 0.55 | 0.50 |
| UCSD34 | $\gamma_1$ | 0.91 | 0.86 | 0.86 | 0.86 |
| | $\gamma_2$ | 0.18 | 0.45 | 0.17 | 0.17 |
| Berkeley13 | $\gamma_1$ | 0.90 | 0.83 | 0.83 | 0.83 |
| | $\gamma_2$ | 0.18 | 0.43 | 0.17 | 0.17 |

directly designed based on $\gamma_2$, the proposed approaches indeed reduce the values of it. In addition, presented $\gamma$ values can also help to explain the ineffectiveness of edge deletion on Pokec networks. Due to the inter-edge sparsity and overly unbalanced structure of these datasets in terms of $|\mathcal{E}^\chi|$ and $|\mathcal{E}^\omega|$ (presented in Table 5 of Appendix A.5), node sampling becomes a better choice than edge deletion/addition to reduce the bias via an augmentation on the graph topology. Table 11 also shows the better effectiveness of node sampling strategy in reducing $\gamma_2$ compared to edge deletion/addition frameworks, which also supports the findings of Table 2, that is, node sampling is an essential step of FairAug on the Pokec networks. Furthermore, $\gamma$ values can also help to explain the random/detrimental effect of node sampling for Facebook networks, which is presented in Table 3. Tables 11 and 12 show that while node sampling is not very effective in reducing $\gamma_1$, it is observed to increase $\gamma_2$ for these networks. This effect results from the graph topologies of Facebook networks having $|\mathcal{S}^\chi| >> |\mathcal{S}^\omega|$. Note that $\gamma_1$ values are not 0 after node sampling due to the limit on the minimum of sampling budgets, see Appendix A.7.

## A.9 CASE STUDIES FOR THE EFFECTS OF PROPOSED AUGMENTATIONS

To demonstrate the effects of the proposed augmentation schemes in a more intuitive manner on different graph structures, we consider two small toy examples. The topology in Figure 1a is inspired by the topology of the Pokec networks (few inter-edges, many more intra-edges, i.e., $|\mathcal{E}^\omega| > |\mathcal{E}^\chi|$), whereas Figure 1b is motivated by the Facebook networks (few nodes with no inter-edges, many more that have at least one, i.e., $|\mathcal{S}^\chi| > |\mathcal{S}^\omega|$). For demonstrative purposes, assume the nodes in both graphs share the same nodal features, with the corresponding feature matrix $\mathbf{X}$ equal to

$$\mathbf{X} = \begin{bmatrix} 0.0 & 0.1 & -0.3 & 0.1 & -0.1 \\ 0.2 & -0.2 & -0.2 & 0.1 & 0.1 \\ 0.1 & 0.2 & 0.0 & 0.2 & 0.0 \\ 0.2 & 0.1 & -0.2 & 0.1 & 0.2 \\ 0.1 & -0.1 & -0.1 & 0.1 & -0.2 \\ -0.1 & -0.1 & 0.3 & -0.2 & 0.3 \\ -0.2 & 0.1 & 0.4 & -0.1 & -0.1 \\ -0.3 & -0.1 & 0.1 & -0.3 & -0.2 \end{bmatrix}.$$

**Feature masking:** For the given $\mathbf{X}$, $\bar{\boldsymbol{\delta}}$ utilized in adaptive feature masking equals to $[0.74, 0.12, 1.00, 0.74, 0.00]$. Therefore, in a feature masking scheme where $40\%$ of the features are masked, the most probable augmented $\tilde{\mathbf{X}}$ becomes[2]

$$\tilde{\mathbf{X}} = \begin{bmatrix} 0.0 & 0.2 & 0.0 & 0.0 & -0.1 \\ 0.0 & 0.1 & 0.0 & 0.1 & -0.1 \\ 0.0 & -0.2 & 0.0 & 0.1 & 0.1 \\ 0.0 & 0.2 & 0.0 & 0.2 & 0.0 \\ 0.0 & 0.1 & 0.0 & 0.1 & 0.2 \\ 0.0 & -0.1 & 0.0 & 0.1 & -0.2 \\ 0.0 & -0.1 & 0.0 & -0.2 & 0.3 \\ 0.0 & 0.1 & 0.0 & -0.1 & -0.1 \\ 0.0 & -0.1 & 0.0 & -0.3 & -0.2 \end{bmatrix}.$$

---

[2] Note that the feature masking process is stochastic. We present the mask with the largest probability of occurrence, for demonstrative purposes.

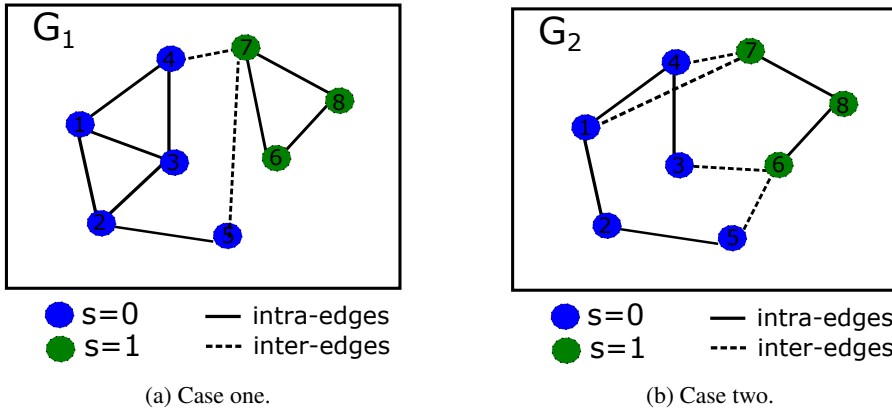

(a) Case one.

(b) Case two.

Figure 1: Toy examples

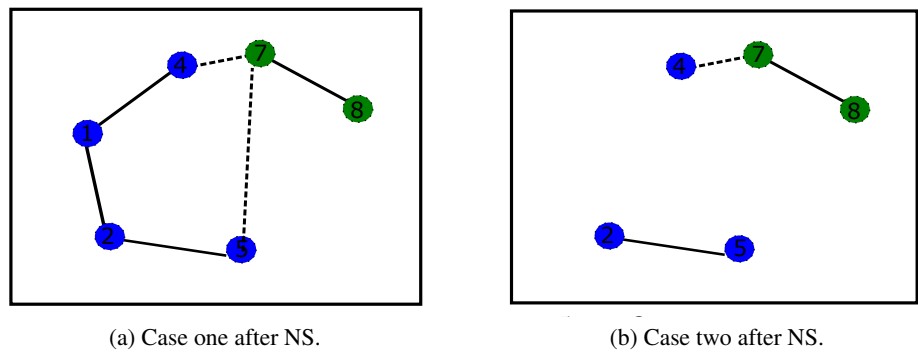

(a) Case one after NS.

(b) Case two after NS.

Figure 2: Augmented graphs via node sampling

For $\tilde{\mathbf{X}}$, $(\boldsymbol{\mu}_1 - \boldsymbol{\mu}_0)$ in equation 15 becomes $[0, -0.05, 0, -0.32, 0]$ where it was $[-0.32, -0.05, 0.43, -0.32, 0]$ for the original graph. Note that this is the best case scenario given the 40% masking budget, with the maximal decrease in $\|\boldsymbol{\mu}_1 - \boldsymbol{\mu}_0\|_1$. As also mentioned in Footnote 2, the actual feature masking process is stochastic. By assigning larger masking probabilities to the features with high deviations between $\mathcal{S}_0$ and $\mathcal{S}_1$, the non-uniform masking strategy proposed herein increases the chances of obtaining the "better" cases with larger reductions in $(\boldsymbol{\mu}_1 - \boldsymbol{\mu}_0)$.

**Node sampling:** The results in Table 3 demonstrate that node sampling becomes effective in graphs where $|\mathcal{S}^\omega| > |\mathcal{S}^\chi|$. Here, we visualize this finding through the toy examples presented in Figures 1a and 1b. Note that the two toy graphs differ in their sets $\mathcal{S}^\omega$ and $\mathcal{S}^\chi$. Specifically, we have $|\mathcal{S}^\omega| > |\mathcal{S}^\chi|$ for Figure 1a and $|\mathcal{S}^\omega| < |\mathcal{S}^\chi|$ for Figure 1b.

Here, the exact $\|\boldsymbol{\epsilon}\|_1 = \|\frac{1}{|\mathcal{S}_1|} \sum_{v_n \in S_1} \mathbf{z}_n - \frac{1}{|\mathcal{S}_0|} \sum_{v_m \in S_0} \mathbf{z}_m\|_1$ values are calculated before and after node sampling for two different graph examples $\mathbf{G}_1$ and $\mathbf{G}_2$. Note that $\mathbf{z}_i = \frac{1}{d_i} \sum_{v_j \in \mathcal{N}(i)} \mathbf{x}_j$, and the original nodal features $\mathbf{X}$ are utilized to calculate the $\mathbf{z}_i$ vectors.

*Case 1:* In the first example graph $\mathbf{G}_1$, $|\mathcal{S}_0^\chi| = 2$, $|\mathcal{S}_0^\omega| = 3$, $|\mathcal{S}_1^\chi| = 1$, and $|\mathcal{S}_1^\omega| = 2$. Therefore, the proposed node sampling scheme selects 2 nodes from the set $\mathcal{S}_0^\omega$, and 1 node from the set $\mathcal{S}_1^\omega$. A possible augmented graph is presented in Figure 2a on which the examination is carried over. While the original $\|\boldsymbol{\epsilon}\|_1$ equals to 0.865 for $\mathbf{G}_1$, $\|\boldsymbol{\epsilon}\|_1$ becomes 0.771 for the output topology of the node sampling provided in Figure 2a, which shows the efficacy of the proposed framework.

*Case 2:* For $\mathbf{G}_2$, $|\mathcal{S}_0^\chi| = 4$, $|\mathcal{S}_0^\omega| = 1$, $|\mathcal{S}_1^\chi| = 2$, and $|\mathcal{S}_1^\omega| = 1$. Therefore, the proposed node sampling scheme selects 2 nodes from the set $\mathcal{S}_0^\chi$ (minimum limit for each set is the 50% of the

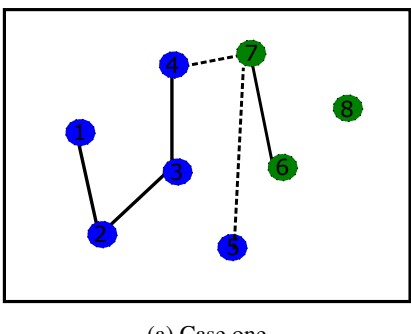 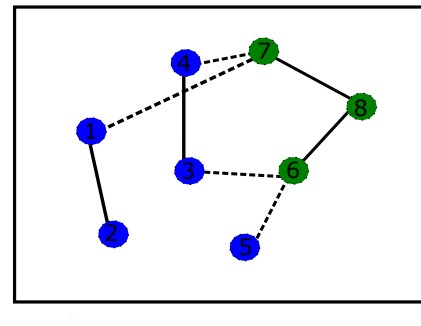

(a) Case one.    (b) Case two.

Figure 3: Augmented graphs via edge deletion

initial group size), and 1 node from the set $\mathcal{S}_1^{\chi}$. Therefore, the presented example in Figure 2b is a potential augmented topology obtained after the proposed node sampling framework. For this case, the original $\|\boldsymbol{\epsilon}\|_1 = 0.558$ which becomes $0.892$ for the augmented graph presented in Figure 2b. This case study clearly illustrates that the proposed node sampling scheme is mainly effective for graphs where $|\mathcal{S}^{\omega}| > |\mathcal{S}^{\chi}|$.

**Edge deletion:** Results in Table 2 show that node sampling is a better choice than edge manipulations for graphs with inter-edge sparsity and $|\mathcal{E}^{\omega}| \gg |\mathcal{E}^{\chi}|$. Again, two case studies based on the graphs provided in Figure 1 are utilized to demonstrate the effects of the proposed adaptive edge deletion scheme. Specifically, the resulted $\|\boldsymbol{\epsilon}\|_1 = \|\frac{1}{|\mathcal{S}_1|} \sum_{v_n \in S_1} \mathbf{z}_n - \frac{1}{|\mathcal{S}_0|} \sum_{v_m \in S_0} \mathbf{z}_m\|_1$ value is examined before and after the application of edge deletion.

*Case 1:* In the first example graph $\mathbf{G}_1$, $|\mathcal{E}^{\chi}| = 2$, $|\mathcal{E}_{\mathcal{S}_0}^{\omega}| = 6$, and $|\mathcal{E}_{\mathcal{S}_1}^{\omega}| = 3$. Therefore, edge deletion probability of the adaptive edge deletion framework is $0.5$ for the edges in $\mathcal{E}_{\mathcal{S}_0}^{\omega}$, and $\mathcal{E}_{\mathcal{S}_0}^{\omega}$ with the hyperparameter selections $\pi = 1$, and $p_{min}^{(e)} = \frac{\pi}{2} = 0.5$. For such a scheme, a possible augmented graph is presented in Figure 3a for demonstrative purposes. While the value of $\|\boldsymbol{\epsilon}\|_1$ is reduced from $0.865$ to $0.792$ with the application of adaptive edge deletion, the graph in Figure 3a also becomes disconnected. For certain real networks, such as the Pokec networks considered in this paper, the imbalance between intra- and inter-edges is considerably larger than the imbalance in the graph in Figure 1a. Therefore, the sparsity resulting from the adaptive edge deletion can be even more severe for such graphs than this toy example. Thus, this case study also confirms that node sampling can indeed be a better choice than edge deletion for graphs with inter-edge sparsity and $|\mathcal{E}^{\omega}| >> |\mathcal{E}^{\chi}|$.

*Case 2:* For $\mathbf{G}_2$, we have $|\mathcal{E}^{\chi}| = 4$, $|\mathcal{E}_{\mathcal{S}_0}^{\omega}| = 4$, and $|\mathcal{E}_{\mathcal{S}_1}^{\omega}| = 2$, which shows a more balanced structure than the graph utilized in Case 1. For the given graph connectivity, edge deletion probability of adaptive edge deletion scheme is $0.5$ for the edges in $\mathcal{E}_{\mathcal{S}_0}^{\omega}$, and $0$ for the edges in $\mathcal{E}_{\mathcal{S}_0}^{\omega}$ together with the hyperparameter selections $\pi = 1$, and $p_{min}^{(e)} = \frac{\pi}{2} = 0.5$. The presented example in Figure 3b is a potential resulting augmented topology obtained with the proposed edge deletion framework, presented for demonstrative purposes. For this outcome, the original $\|\boldsymbol{\epsilon}\|_1$ is decreased from $0.558$ to $0.497$ for the augmented graph presented in Figure 3b, which demonstrates the effectiveness of the proposed augmentation for the input graph in Case 2.

**Edge Addition:** We also present the effects of the proposed edge addition scheme on the toy examples for a better illustration of the augmentation. Again, the exact $\|\boldsymbol{\epsilon}\|_1 = \|\frac{1}{|\mathcal{S}_1|} \sum_{v_n \in S_1} \mathbf{z}_n - \frac{1}{|\mathcal{S}_0|} \sum_{v_m \in S_0} \mathbf{z}_m\|_1$ values are calculated before and after edge addition for two different graph examples $\mathbf{G}_1$ and $\mathbf{G}_2$.

*Case 1:* For $\mathbf{G}_1$, as $|\mathcal{E}^{\omega}| = 9$ and $|\mathcal{E}^{\chi}| = 2$, new seven artificial inter-edges are generated for the proposed edge addition scheme. For such a scheme, a possible augmented graph is presented in Figure 4a, where $\|\boldsymbol{\epsilon}\|_1$ is reduced from $0.865$ to $0.111$ with the application of edge addition.

*Case 2:* For $\mathbf{G}_2$, as $|\mathcal{E}^{\omega}| = 6$ and $|\mathcal{E}^{\chi}| = 4$, therefore two artificial inter-edges should be created for the proposed edge addition scheme. A possible augmented graph resulting from the proposed edge

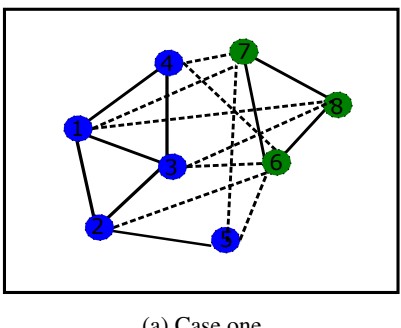 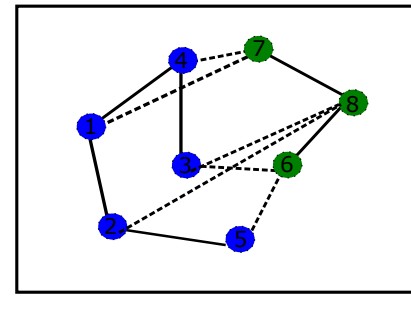

(a) Case one.                                    (b) Case two.

Figure 4: Augmented graphs via edge addition

addition is presented in Figure 4b. For the presented example in Figure 4b, edge addition can lower the value of $\|\epsilon\|_1$ from $0.558$ to $0.293$.

## A.10 OPTIMAL TOPOLOGY AUGMENTATION SCHEMES FOR $\gamma_1$ AND $\gamma_2$

While Algorithm 1 can make $\gamma_1 = 0$ for graphs satisfying $|\mathcal{S}^\omega| \geq |\mathcal{S}^\chi|$, as Table 12 demonstrates, the obtained $\gamma_1$ values after node sampling are not exactly zero due to the limit on the number of nodes that can be sampled. If $|\mathcal{S}^\chi| \geq |\mathcal{S}^\omega|$, the minimum number of nodes that will be sampled for each set is $50\%$ of the initial group size, otherwise the limit is $25\%$ of the initial group size. Such limits are employed to prevent the generation of excessively small subgraphs, which can prevent proper learning and create stability issues. Furthermore, proposed global edge deletion/addition frameworks cannot directly make $\gamma_2$ zero, as the reduction of $\gamma_2$ to zero requires a node-wise consideration. Since such an independent consideration of each node can incur a significant computational complexity for large graphs, we stay within the realm of global schemes that can still help to reduce the value of $\gamma_2$. However, the examination of the optimal schemes that can make $\gamma_1 \gamma_2$ zero helps provide important insights and justification for the proposed augmentations.

Motivated by this, we examined "optimal" strategies which can drive $\gamma_1$ or $\gamma_2$ to zero. The corresponding node sampling strategy copes with each node independently by deleting/adding edges so that $d_i^\omega = d_i^\chi, \forall v_i \in \mathcal{V}$. Different from the proposed global edge manipulation scheme in Section 3.3.3, such node-level edge deletion/addition frameworks can change the value of $\gamma_1$ and henceforth interfere with the effects of the node sampling step. Additionally, the employment of edge deletion for the graph makes the utilization of edge addition unnecessary and vice versa, as both schemes seek to achieve the same goal, i.e., $d_i^\omega = d_i^\chi, \forall v_i \in \mathcal{V}$. Therefore, only the "optimal" edge deletion or addition scheme is employed. We would like to note that the investigation is carried out for Pokec networks, as $|\mathcal{S}^\omega| \leq |\mathcal{S}^\chi|$ for Facebook networks, which makes the design of an optimal node sampling scheme challenging due to the unpredictable increase in $\mathcal{S}^\omega$ and the unpredictable decrease in $\mathcal{S}^\chi$, when sampling from the set $\mathcal{S}^\chi$ (some of the excluded nodes and edges may be the only inter-edges for the nodes in the sampled graph, resulting in a further increase in $\mathcal{S}^\omega$ and further decrease in $\mathcal{S}^\chi$).

Table 13: Effects of optimal augmentations on $\gamma_1$

|         | Original | Node Sampling | Edge Deletion | Edge Addition |
|---------|----------|---------------|---------------|---------------|
| Pokec-z | 0.66     | 0.00          | 0.67          | 1.00          |
| Pokec-n | 0.67     | 0.00          | 0.67          | 1.00          |

Table 14: Effects of optimal augmentations on $\gamma_2$

|         | Original | Node Sampling | Edge Deletion | Edge Addition |
|---------|----------|---------------|---------------|---------------|
| Pokec-z | 0.90     | 0.48          | 0.00          | 0.00          |
| Pokec-n | 0.91     | 0.46          | 0.00          | 0.00          |

Table 15: Effects of each step in Node Sampling + Edge Deletion

|  |  | Original Graph | Node Sampling | Edge Deletion |
|---|---|---|---|---|
| Pokec-z | $\gamma_1$ | 0.66 | 0.00 | 0.08 |
|  | $\gamma_2$ | 0.90 | 0.48 | 0.00 |
| Pokec-n | $\gamma_1$ | 0.67 | 0.00 | 0.11 |
|  | $\gamma_2$ | 0.91 | 0.46 | 0.00 |

Table 16: Effects of each step in Node Sampling + Edge Addition

|  |  | Original Graph | Node Sampling | Edge Addition |
|---|---|---|---|---|
| Pokec-z | $\gamma_1$ | 0.66 | 0.00 | 0.73 |
|  | $\gamma_2$ | 0.90 | 0.48 | 0.00 |
| Pokec-n | $\gamma_1$ | 0.67 | 0.00 | 0.70 |
|  | $\gamma_2$ | 0.91 | 0.46 | 0.00 |

Tables 13-16 demonstrate that the utilized node sampling, edge deletion/addition frameworks are optimal in the sense that they reduce $\gamma_1$ and $\gamma_2$ to zero, respectively. However, presented $\gamma_1$ and $\gamma_2$ values also show the interference of optimal edge manipulations on $\gamma_1$, as $\gamma_1$ increases after the application of edge deletion/addition following node sampling. Specifically, edge addition has a great impact on it.

