# OpenReview forum: "Fair Node Representation Learning via Adaptive Data Augmentation"
_ICLR.cc/2022/Conference — ICLR 2022 Submitted_

### Official Review · Reviewer_xMz4 · 2021-11-01

**Correctness:** 2
**Technical Novelty And Significance:** 3
**Empirical Novelty And Significance:** 3
**Recommendation:** 3
**Confidence:** 5

**Main Review:**

Several main concerns should be well-addressed by the authors.

1. In the Introduction, the motivation is not clear.
Why the "augmentation method" can well address this problem? It seems that the authors find that "augmentation has not been employed for fairness learning on graph, so we do".
Besides, many related studies have been proposed for fair node representations. Can they deal with this problem? The authors do not illustrate how and why the proposed model is better than theirs.

2. In the section of Related Work, the related studies are not sufficiently surveyed, and several recent related studies such as [1,2,3] are missed.
Besides, in the last sentence of page 2, the authors claim "it is not adaptive to the graph structure". What does it mean? And why? The authors should give more explanations.
Furthermore, the authors do not provide the drawbacks of these related studies compared to the proposed model.

3. At the end of page 3, the authors give an equation to depict the calculation of correlation between the sensitive attributes and the aggregated representations.
However, this equation is a bit confusing. In particular, in the numerator of this equation, what does "n" in the sum operator mean?
Besides, in the following paragraph, why "it is assumed that these distributions have the same maximal deviation ∆"?

4. In the first paragraph of section 3.3.1, the authors claim that "for the first layer of GNN without losing significant information that the features provide". However, there is no content to demonstrate the proposed model can fulfill the goal of "without losing significant information". Both the features and the structures are modified to train the GNN model (such as by feature mask, as well as edge addition and deletion), thus much important information could be lost. So how to guarantee this claim?

5. In the first paragraph of section 3.3.3, there are two confusing equations |E_x|<=|E_o| and |E_x|>=|E_o|.

6. With the proposed model, some augmented graphs can be achieved. However, the authors do not give the details that how to use these graphs for contrastive training. To maximize their agreement?
Generally, contrastive learning usually aims to learn representations with better generalizability, transferability and robustness. So why this kind of contrastive learning, though with the proposed augmentations, can address the fairness issue?

7. In experiments, the authors employ citation networks Cora, Citeseer and Pubmed for fairness evaluation, and utilize the category of the articles as their sensitive attributes. However, this kind of utilization is not reasonable. Sensitive attributes are the inherent attributes of instances, and can involve some kind of discrimination. However, here the category of articles is not an inherent attribute but is summarized by humans. Would they involve any discrimination in the articles? I don't think so.
For the fairness evaluation of node representations, the authors employ two widely used metrics, namely statistical parity and equal opportunity. For example, for nodes with different genders, the gender might influence their labels. However, for the fairness evaluation of edges, the proposed metrics are not well-motivated. In particular, for edges belonging to different groups (E^X or E^W), do this kind of sensitive attributes (i.e., the groups) influence their existence? Besides, why the generation of fair node representations, can generate fair edges?

8. In experiments, most of the fairness learning approaches are not employed as the baselines for comparison, such as the related studies in section 2, as well as [1,2]. Thus, the experiments are not convincing.


[1] Compositional fairness constraints for graph embeddings. ICML 2019

[2] Debayes: a bayesian method for debiasing network embeddings. ICML 2020

[3] Debiasing knowledge graph embeddings. EMNLP 2020

**Summary Of The Paper:**

This paper investigates the problem of fairness learning on graphs. To address this, the authors first give a theoretical analysis on the sources of bias in node representations achieved by GNNs, then propose several data augmentation methods for fair node representation learning. Experiments on several datasets demonstrate the effectiveness of the proposed model.

**Summary Of The Review:**

1. The motivation of the paper is not clear enough.
2. Several claims are not well-demonstrated.
3. The experimental settings are not quite reasonable, and the experiments are not convincing enough.

---

> ### Author Response · Authors · 2021-11-20
> **Response to Reviewer xMz4: 8**
>
> __Q8:__ In experiments, most of the fairness learning approaches are not employed as the baselines for comparison, such as the related studies in section 2, as well as [1,2]. Thus, the experiments are not convincing.
>
> __R8:__ While we acknowledge that there are many fairness-aware learning approaches in the literature including [1,2] mentioned by the Reviewer, we did not include all of them, but instead clarified our novelty in the related work (Section II) and compared our framework with some of the most relevant and recent works.
>
> Specifically, we compared our study with [R5], which is more recent than [1,2], and it also utilizes a fairness-aware graph augmentation in the framework of graph contrastive learning, which to the best of our knowledge, is the most recent and closest to our work.
>
> [R5] Towards a Unified Framework for Fair and Stable Graph Representation Learning, UAI, 2021
>
> In addition, as we also mentioned in our response to the Reviewer’s Comment 6, the contrastive learning application of the proposed scheme is built upon GRACE [R3]. This makes GRACE a natural baseline for our study. To this end, the comparison with GRACE [R3] is included to evaluate the efficacy of the proposed augmentations in terms of fairness.
>
> [R3] Deep Graph Contrastive Representation Learning, ICML 2020
>
> In addition to these studies, we also compared with another study considering fair node representation learning FairWalk [R2] alongside its baseline study, Node2Vec [R8].
>
> [R2] Fairwalk: Towards Fair Graph Embedding, IJCAI, 2019
>
> [R8] Node2vec: Scalable feature learning for networks, ACM SIGKDD, 2016
>
>
> ***
>
>
> FInally, we would like to thank the Reviewer for the comments and raised remarks once again.

---

> ### Author Response · Authors · 2021-11-20
> **Response to Reviewer xMz4: 7**
>
> __Q7a:__ In experiments, the authors employ citation networks Cora, Citeseer and Pubmed for fairness evaluation, and utilize the category of the articles as their sensitive attributes. However, this kind of utilization is not reasonable. Sensitive attributes are the inherent attributes of instances, and can involve some kind of discrimination. However, here the category of articles is not an inherent attribute but is summarized by humans. Would they involve any discrimination in the articles? I don't think so.
>
> __Q7b:__ For the fairness evaluation of node representations, the authors employ two widely used metrics, namely statistical parity and equal opportunity. For example, for nodes with different genders, the gender might influence their labels. However, for the fairness evaluation of edges, the proposed metrics are not well-motivated. In particular, for edges belonging to different groups (E^X or E^W), do this kind of sensitive attributes (i.e., the groups) influence their existence?
>
> __Q7c:__ Besides, why the generation of fair node representations, can generate fair edges?
> ***
>
> __R7:__ Firstly, we would like to clarify that our work is not the first one that utilizes citation networks for the evaluation of fairness-aware link prediction: Citation networks are used by considering the category of the article as the sensitive attribute in [R7, R9]. Furthermore, the same or similar fairness measures used for link prediction in this manuscript (statistical parity $\Delta_{S P}$ and equal opportunity $\Delta_{E O}$) are also employed in [R2, R7, R9].
>
> ***
>
> __R7-a:__ We understand the concern regarding the utilization of the category of the articles as their sensitive attributes from a societal point of view. However, our study is not the first one that considers article category as the “sensitive attribute” in citation networks; such experimental setting has also been used in previous works, see e.g., [R7]. Our understanding is that the fairness problem considered here can be (mathematically) regarded as the problem of decorrelating the system output from a particular attribute associated with the nodes. While we agree with the Reviewer that the sensitive attributes should have its societal implication or impact, the problem is decoupled from its societal implications from an algorithmic point of view.
>
> ***
>
> __R7-b:__
>
> First, we again want to clarify that the same or similar fairness measures that are used in our study for link prediction have been utilized in the literature for fairness aware link prediction [R2,R7,R9]. Such measures are motivated by the fact that users tend to connect to other users with the same sensitive attributes, e.g. gender, religion, political view in social networks. A data-driven recommendation algorithm can capture the dependency between the link formation and sameness of the sensitive attributes, and can further propagate such dependency in the recommendations, leading to a segregation in the sensitive attribute. In this context, we note that the fairness metrics utilized in link prediction measure the discrepancy of the outcomes for ​​inter-edges ($\mathcal E^\chi$) and intra-edges ($\mathcal E^\omega$) in a link prediction task. Therefore, the utilized fairness measures actually employ the sameness of the sensitive attributes as sensitive groups (i.e. the similar predictions for the existence of inter- and intra- edges).
>
> Similar to social networks, sensitive attributes utilized in the citation networks can indeed influence the existence of edges. If the category of the article is considered as the sensitive attribute, Table 6 shows that the number of intra-edges is larger than the number of inter-edges for citation networks. A recommender system trained on such data would be biased to recommend articles within the same group (prediction of intra-links with higher probabilities), implying that the sameness of the categories of the articles can influence the predictions for the existence of an edge.
>
>
> ***
>
>
> __R7-c:__ We would like to clarify that we do not aim for fair edges. The idea of our work is to generate nodal representations whose correlations with the sensitive attributes are reduced. Such “de-correlated” representations can enhance the fairness metrics for ensuing tasks, such as node classification and link prediction, in the sense that the results are also “de-correlated” with the sensitive attributes. As mentioned before, the bias in link prediction occurs by assigning higher prediction probabilities to intra-edges, which is resulted from the utilization of sensitive information. The de-correlated representations can lower such utilization, leading to an improvement in fairness metrics.
>
>
> ***
>
> [R2] Fairwalk: Towards Fair Graph Embedding, IJCAI, 2019
>
> [R7] On Dyadic Fairness: Exploring and Mitigating Bias in Graph Connections, ICLR, 2021
>
> [R9] Biased Edge Dropout for Enhancing Fairness in Graph Representation Learning, Arxiv, 2021

---

> ### Author Response · Authors · 2021-11-20
> **Response to Reviewer xMz4: 4, 5 and 6**
>
> __Q4:__ In the first paragraph of section 3.3.1, the authors claim that "for the first layer of GNN without losing significant information that the features provide". However, there is no content to demonstrate the proposed model can fulfill the goal of "without losing significant information". Both the features and the structures are modified to train the GNN model (such as by feature mask, as well as edge addition and deletion), thus much important information could be lost. So how to guarantee this claim?
>
> __R4:__ We understand the confusion, and we have changed the sentence to the following in the revised manuscript:
>
> “Note that $||\boldsymbol \delta||_1$ in equation 1 is minimized when all nodal features are the same (i.e., all nodal features masked/zeroed out). However, this would result in the loss of all information coming from nodal features. Motivated by this, the proposed scheme aims to improve uniform feature masking in terms of reducing $||\boldsymbol \delta||_1$ for a given masking budget (a total amount of nodal features to be masked in expectation).”
>
> We would like to emphasize that the more features are masked/zeroed out, the smaller $||\boldsymbol \delta||_1$ would be. Masking all features trivially achieves the minimal $||\boldsymbol \delta||_1$, since it zeroes out all nodal features. However, as this solution erases all nodal features, it loses all information provided by them. In this context, there is a trade-off between the amount of nodal features left unmasked and the value of $||\boldsymbol \delta||_1$. To this end, our proposed feature masking aims to provide a better yield in terms of $||\boldsymbol \delta||_1$ reduction compared to uniform feature masking (i.e., the baseline used in graph contrastive learning [R3, R5]), given a fixed “masking budget” (see Proposition 1).
>
>
> ***
>
> __Q5:__ In the first paragraph of section 3.3.3, there are two confusing equations |E_x|<=|E_o| and |E_x|>=|E_o|.
>
> __R5:__ We thank the Reviewer for pointing this out. This was a typo in the original manuscript, which we fixed in the revised version (which now writes $|\mathcal E^\chi| \leq |\mathcal E^\omega|$  and $|\mathcal E^\chi| > |\mathcal E^\omega|$, respectively).
>
>
> ***
>
>
> __Q6:__ With the proposed model, some augmented graphs can be achieved. However, the authors do not give the details that how to use these graphs for contrastive training. To maximize their agreement? Generally, contrastive learning usually aims to learn representations with better generalizability, transferability and robustness. So why this kind of contrastive learning, though with the proposed augmentations, can address the fairness issue?
>
> __R6:__ The implementation of the utilized contrastive learning scheme was presented in detail in Appendix A.6. Specifically, after the creation of two graph views through graph data augmentations, the employed contrastive learning framework maximizes the agreement of the nodal representations generated with these two views [R3].
>
> Furthermore, we would like to clarify that our fairness analysis is based on GNN-based learning, not specifically on contrastive learning. As the training of the utilized contrastive learning framework is GNN-based, the source of the fairness results from the employment of GNN, not particularly contrastive learning.
>
> We would like to emphasize that the proposed augmentations would be compatible with other implementations of contrastive training other than GRACE as well, as long as graph augmentations are utilized within the training process.
>
> In addition, our proposed framework is not limited to its application on contrastive learning (see also our response to Q1-b). Instead, contrastive learning is just one of the possible use cases of the proposed augmentations. In addition to its application in contrastive learning, the proposed augmentation schemes can also be considered as pre-processing tools before training for GNN-based learning. For example, such a use case can be the employment of the proposed edge deletion (ED) scheme as a fairness-aware edge dropout method. As can also be confirmed from Table 4, such a “Fair ED” strategy indeed improves fairness over a fairness-unaware edge dropout.
>
> ***
>
> [R3] Deep Graph Contrastive Representation Learning, ICML 2020
>
> [R5] Towards a Unified Framework for Fair and Stable Graph Representation Learning, UAI, 2021

---

> ### Author Response · Authors · 2021-11-20
> **Response to Reviewer xMz4: 2 and 3**
>
> __Q2:__ In the section of Related Work, the related studies are not sufficiently surveyed, and several recent related studies such as [1,2,3] are missed. Besides, in the last sentence of page 2, the authors claim "it is not adaptive to the graph structure". What does it mean? And why? The authors should give more explanations. Furthermore, the authors do not provide the drawbacks of these related studies compared to the proposed model.
>
> [1] Compositional fairness constraints for graph embeddings. ICML 2019
>
> [2] Debayes: a bayesian method for debiasing network embeddings. ICML 2020
>
> [3] Debiasing knowledge graph embeddings. EMNLP 2020
>
>
> __R2:__ We would like to thank the Reviewer for raising this point. We have cited the papers mentioned and added their corresponding text, under “Fairness-aware learning on graphs'' in Section 2 (Related Work).
>
> Throughout the paper, “adaptiveness to graph structure” refers to the dependence of the proposed frameworks to the graph topology (a similar definition can also be made for nodal features). A similar definition is also utilized in [R4] as well (one of the baselines in our paper), which motivated our definition herein. In this context, [R9] is “not adaptive to the graph structure” in the sense that the deletion probabilities of inter-/intra-edges do not take the graph structure into account. In contrast, the deletion probabilities designed in our paper do consider the graph structure (see equation (7)). Following the Reviewer’s suggestion, we clarified “adaptiveness to graph structure” within “Fairness-aware learning on graphs” under Section 2 (Related Work), by adding the following text:
>
> “A biased edge dropout scheme is proposed in [R9] to improve fairness. However, the scheme therein is not adaptive to the graph structure (the parameters of the framework are independent of the input graph topology).”
>
> In the revised manuscript, we considerably revised the “Fairness-aware learning on graphs” subsection following the Reviewer’s comments, and further highlighted the novelties and advantages of our work over the prior art. For further explanation, we refer the Reviewer to our response to Q1-b.
>
>
> ***
>
>
> __Q3:__ At the end of page 3, the authors give an equation to depict the calculation of correlation between the sensitive attributes and the aggregated representations. However, this equation is a bit confusing. In particular, in the numerator of this equation, what does "n" in the sum operator mean? Besides, in the following paragraph, why "it is assumed that these distributions have the same maximal deviation $\Delta$"?
>
> __R3:__ The  “n” in the sum operator was a typo, and it should have been “N”, the number of nodes. We fixed it in the revised manuscript. We thank the Reviewer for this keen observation.
>
> The assumption that these distributions have the same maximal deviation is for notational convenience and without loss of generality. To address this comment, we have revised the assumptions such that the representation from the two groups have finite maximal deviations $\Delta_{0}$ and $\Delta_{1}$, and Theorem 1 still holds with $\Delta := \max (\Delta_{0}, \Delta_{1})$. Please see Theorem 1 and Appendix A.1 for more details.
>
>
> ***
>
> [R4] Graph Contrastive Learning with Adaptive Augmentation, WWW, 2021
>
> [R9] Biased Edge Dropout for Enhancing Fairness in Graph Representation Learning, Arxiv, 2021

---

> ### Author Response · Authors · 2021-11-20
> **Response to Reviewer xMz4: 1**
>
> We thank the Reviewer for the comments. Below, we responded to the Reviewer’s concerns in our point-to-point replies.
>
> ***
>
> __Q-1a:__ In the Introduction, the motivation is not clear. Why the "augmentation method" can well address this problem? It seems that the authors find that "augmentation has not been employed for fairness learning on graph, so we do".
>
> __R-1a:__ We respectfully disagree that the motivation of this work is "augmentation has not been employed for fairness learning on graph, so we do”. In fact, augmentation has been employed in [R5] to enhance fairness, which is included as a benchmark in our numerical results (as NIFTY). In addition, compared with [R5], the novelty of this work (as stated in the contribution c1 in introduction) is that the augmentation design is inspired by theoretical analysis regarding the sources of the bias, which provides important insights and justification for the effectiveness of the augmentation design.
>
> Regarding the motivation of this work,  the starting point is not directly tackling the fairness problem via graph data augmentation. Instead, the goal is to theoretically explain the sources of bias and in turn improve fairness in graph representation learning by reducing the corresponding terms that lead to bias. To achieve this goal, we analyzed the correlation between sensitive attributes (__s__) and the aggregated representations (__Z__), which is described by the parameter $\boldsymbol{\rho}$ in the manuscript. The bound provided in Theorem 1 upper bounds $||\boldsymbol \rho||_1$ in terms of the system parameters (e.g. $|\mathcal E^\chi|, |\mathcal E^\omega|, |\mathcal S_0^\chi|, |\mathcal S_1^\omega|$, etc.). Motivated by the findings in Theorem 1, this work aims to reduce the terms in the upper bound (i.e., $||\boldsymbol \delta||_1$, $\gamma_1$, and $\gamma_2$), which can be achieved by graph data augmentation schemes developed in the paper. In particular, node sampling addresses $\gamma_1$, edge manipulations $\gamma_2$, and adaptive feature masking $||\boldsymbol \delta||_1$. Experimental results presented in Appendix A.8 also show the effectiveness of our scheme in reducing the corresponding terms.
>
> To address this comment, we have revised the introduction to make the motivation more clear.
>
>
> ***
>
>
> __Q1-b:__ Besides, many related studies have been proposed for fair node representations. Can they deal with this problem? The authors do not illustrate how and why the proposed model is better than theirs.
>
> __R1-b:__ Thank you for the question. To clarify the advantage and novelty of the proposed framework compared with existing works, we have revised the corresponding paragraph in Related Work (Section 2). For more details, please check “Fairness-aware learning on graphs” in Section 2.
>
> In a nutshell, the novelty of our work can be summarized as follows:
>
> Unlike the majority of prior works, the proposed framework is built upon theoretical analysis explaining the sources of bias (see Theorem 1), and the augmentation schemes are inspired by the analysis in order to reduce the terms in the bound. We also theoretically proved the efficacy of the feature masking scheme in Proposition 1 (see Appendix A.2).
>
> Similar to prior art, the proposed framework can be utilized within the learning process to mitigate bias by modifying the learned model, serving as an “in-processing fairness strategy” [R6] (e.g. in the learning framework of contrastive learning). However, unlike most prior art, the introduced schemes can also be regarded as "pre-processing" tools that alter the graph structure before the learning algorithm is initiated. This implies that the proposed theoretical analysis and augmentation schemes can benefit a plethora of GNN-based learning schemes, and suggests a desirable versatility.
>
>
> ***
>
> [R5] Towards a Unified Framework for Fair and Stable Graph Representation Learning, UAI, 2021
>
> [R6] A Survey on Bias and Fairness in Machine Learning, ACM Computing Surveys, 2021

---

### Official Review · Reviewer_FSrq · 2021-11-03

**Correctness:** 3
**Technical Novelty And Significance:** 3
**Empirical Novelty And Significance:** 3
**Recommendation:** 6
**Confidence:** 3

**Main Review:**


Strengths:
1. The proposed method improves fairness metrics on node classification and link prediction while maintaining decent performance on accuracy.
2. The proposed method is simple to implement and justified by the theoretical analysis on the upper bound.

Weaknesses:
1. The hyperparameters for data augmentation provided in appendix (A.7) seem to be a bit like magic numbers. How do the author determine to use these values rather than other values? Also, the sensitivity of the proposed methods w.r.t the hyperparameters should be investigated.
2. The effect of each data augmentation technique might interfere with each other. How to specify the right amount of augmentation for each type is not well studied.
3. Some case study on the real-world dataset or even toy dataset can be helpful for readers to better understand these data augmentation techniques in a more intuitive manner.



**Summary Of The Paper:**

This paper studies the bias issue in the representation learned by GNN-based models. Some theoretical analysis is presented, and based on this analysis they propose several data augmentation techniques, such as feature masking, edge addition/deletion and node sampling. Experimental results demonstrate the effectiveness of these augmentation on multiple real-world graphs.

**Summary Of The Review:**

Overall the proposed technique following the analysis on the theoretical upper bound, is simple and effective in improving fairness metrics. Experiments could be made stronger though, such as by including sensitivity analysis on the hyperparameters.

---

> ### Author Response · Authors · 2021-11-20
> **Response to Reviewer FSrq: 2 and 3**
>
>
>
> __Q2:__ The effect of each data augmentation technique might interfere with each other. How to specify the right amount of augmentation for each type is not well studied.
>
> __R2:__ Thank you for raising this point. We agree with the reviewer that the effects of each data augmentation technique might indeed interfere with each other. Motivated by this, FairAug is designed to minimize the interaction between different augmentations. While the proposed node sampling modifies the distribution of edges hence affecting $\gamma_2$, the introduced edge manipulations do not influence the value of $\gamma_1$ (thus they do not interfere with node sampling in terms of the value of $\gamma_1$). Therefore, ‘FairAug’ first employs node sampling, and then uses edge manipulations to reduce $\gamma_2$ without interfering $\gamma_1$. Note that the feature masking is applied at last, since it only affects $||\boldsymbol \delta||_1$ but not $\gamma_1$ and $\gamma_2$. This way, we reduce the interference between different augmentation techniques (also see Remark 2 of the manuscript). In the manuscript, Appendix A.8 shows the sequential and independent effects of the proposed augmentations on the values of $\gamma_1$ and $\gamma_2$. Specifically, Tables 11 and 12 provide the effects of different augmentations on the terms $\gamma_1$ and $\gamma_2$.
>
> Furthermore, our feature masking and edge deletion frameworks have parameters to specify a certain amount of augmentation ($\alpha$ and $\pi$). These hyperparameters can be optimized (a grid search over a cross-validation set) with respect to the input graph to choose the right amount of augmentation. For further explanation regarding hyperparameter tuning for $\alpha$ and $\pi$, we refer the Reviewer to our answer to Comment 1, as well as the sensitivity analysis provided in Appendix A.7 of the revised manuscript.
>
>
> ***
>
>
> __Q3:__ Some case study on the real-world dataset or even toy dataset can be helpful for readers to better understand these data augmentation techniques in a more intuitive manner.
>
>
> __R3:__ We would like to thank the Reviewer for the constructive suggestion. To this end, in addition to the experiments in the original manuscript (which are all on real-world graph datasets), we provided two more toy examples to show the effects of data augmentation techniques in a more intuitive manner in Appendix A.9 of the revised manuscript. In the added appendix, we provided two toy graph models (with different topologies in terms of the balances of $\mathcal E^\chi / \mathcal E^\omega$ and $\mathcal S^\chi / \mathcal S^\omega$) to better illustrate the effects of the proposed graph augmentation schemes on the graph structure and nodal features, and to help clarify the findings in experimental results. Specifically, we presented the influence of nodal features on the term ($\boldsymbol \mu_0-\boldsymbol \mu_1$) and the effects of the proposed graph topology augmentations on $||\boldsymbol \epsilon||_1$ (Equation 22 in Appendix A.1).
>
> We would like to thank the Reviewer for this comment, as we believe addressing it has improved the presentation quality of the manuscript.
>
>
> ***
>
> We would like to present our thanks to the Reviewer once more. We believe addressing their comments significantly improved the paper, especially with the introduction of Appendix A.9, and the thorough revision of Appendix A.7.

---

> ### Author Response · Authors · 2021-11-20
> **Response to Reviewer FSrq: 1**
>
> We would like to sincerely thank the Reviewer for the raised points. We are pleased to have addressed all points raised by the Reviewer, and presented our responses below.
>
> ***
>
> __Q1:__ The hyperparameters for data augmentation provided in appendix (A.7) seem to be a bit like magic numbers. How do the author determine to use these values rather than other values? Also, the sensitivity of the proposed methods w.r.t the hyperparameters should be investigated.
>
> __R1:__ We would like to thank the Reviewer for raising this comment. In order to clarify the selection process of hyperparameters, Appendix A.7 is thoroughly modified in the revised manuscript. Following the Reviewer’s suggestion, in the revised Appendix A.7, we included sensitivity analyses for hyperparameters $\alpha$ (feature masking) and $\pi$ (edge deletion) which are the only two hyperparameters to be tuned in the proposed strategies. Specifically, these sensitivity examinations are presented in Tables 7, 8, 9, and 10 in Appendix A.7. The results for $\alpha$ demonstrate that the original results presented in Tables 1 and 3 can actually be improved via tuning this hyperparameter, and all the examined values lead to superior fairness performances compared to our baseline GRACE [R3]. The results for the sensitivity of $\pi$ on Pokec networks show that the selection of $\pi=1$ is a good choice, as it results in the best fairness performance compared to other examined values. However, again, all considered values for $\pi$ are observed to obtain better fairness results compared to our baseline GRACE [R3] with similar classification accuracies. For Facebook networks, the sensitivity analysis of $\pi$ shows that the presented fairness results in Table 3 for $\pi=0.8$ can be slightly improved by tuning this hyperparameter, as the performances are generally stable for different $\pi$ values.
>
> In addition, it is further clarified in the revised Appendix A.7 that all experimental settings of contrastive learning are kept the same as used in our baseline GRACE [R3]. The implementations of node classification and link prediction experiments are again kept the same as our baselines for fair comparison.
>
> To provide further insight, we recall from the above paragraph that our proposed augmentation frameworks have only two hyperparameters to be tuned: $\alpha$ in feature masking, $\pi$ in adaptive edge deletion. The selection of the parameters are clarified as follows:
>
> $\alpha$: The average feature masking probability ($\alpha$) in our scheme is kept the same as all baselines  GCA [R4], GRACE [R3], and NIFTY [R5] which also utilize feature masking schemes for fair comparison.  We did not tune the $\alpha$ value, instead the default hyperparameter setting provided in GCA [R4] is used, which is also clarified in the revised Appendix A.7.
>
> $\pi$: $\pi=1$ for Pokec and citation networks. We consider $\pi$ as 1 in these networks, as such a value is the most natural selection to delete only intra-edges where inter-edges are already scarce for unbalanced networks in terms of inter- and intra-edges (e.g. Pokec, and citation networks). For Facebook graphs, one of the sensitive groups has a very balanced graph topology in terms of inter- and intra-edges. Therefore, for $\pi=1$, the proposed edge deletion scheme in equation 7 results in no edge deletion in one of the sensitive groups (as the ratio of cardinalities of inter- and intra-edges are utilized) and declines the stochastic nature of the framework. Motivated by this, we selected $\pi=0.8$, a value less than 1, but not too small so that most of the inter-edges will be protected in the resulting graph. Note that, as the sensitivity analysis in Table 10 demonstrates that $0.8$ is not an optimal selection, and the presented results in Table 3 can be further improved via a grid search. However, the variation is rather low on Facebook networks for different $\pi$ values, and $\pi=0.8-1$ can provide decent fairness results for balanced networks in terms of inter- and intra-edges, if tuning is preferred to be skipped.
>
> We refer the Reviewer to the revised version of the Appendix A.7, as it further explains the selection of hyperparameter values.
>
>
> ***
>
> [R3] Deep Graph Contrastive Representation Learning, ICML 2020
>
> [R4] Graph Contrastive Learning with Adaptive Augmentation, WWW, 2021
>
> [R5] Towards a Unified Framework for Fair and Stable Graph Representation Learning, UAI, 2021

---

### Official Review · Reviewer_LHQM · 2021-11-03

**Correctness:** 4
**Technical Novelty And Significance:** 4
**Empirical Novelty And Significance:** 4
**Recommendation:** 8
**Confidence:** 2

**Main Review:**

I want to preface my review by stating that I am not very familiar with graph neural networks, and will thus be mostly evaluating the “fairness aspect” of the paper.

**Strengths**
- `The problem setting is relevant.` Fair representation learning for tabular data has been studied extensively in the literature. However, fair representation learning has not been considered yet for graphs, which is the problem that this paper investigates. As the paper empirically demonstrates, prior work on graph representation learning typically incurs group fairness violations, which need to be addressed.
- `The method is novel and well-motivated.` The paper theoretically analyses the correlation between sensitive attributes and the graph representation and uses the analysis to propose different graph augmentation methods to reduce (intrinsic) bias of the representations.
- `The paper is clearly written.` Even as an outsider of the field of graph neural networks I was able to follow the paper as the ideas are well-motivated and clearly presented.

**Weaknesses**
- `The ablation study yields inconclusive results.` Depending on the task, different ablated versions of the proposed method (e.g., without edge deletion) significantly outperform the full method, which is not ideal. In my opinion, it would be better to tailor the approach to the different tasks (e.g., by removing the obsolete parts), rather than leaving the tuning to the practitioners interested in applying the method in practice.

**Questions**
- How do the different graph augmentation strategies interact? The paper conducts an ablation study to compare the performance of different augmentation strategies, but it would be interesting to see if any theoretical conclusions can be made.
- The equation for $\gamma_2$ in theorem 1 and section 3.3.3 are inconsistent. Which is the correct one?
- Proposition 1 shows that in expectation, the proposed adaptive feature masking scheme results in a lower distribution difference compared to uniform feature masking. However, the uniform feature masking is not the baseline we care about (I think). Instead, we would like to know how the proposed masking scheme affects the distributions compared to no masking at all. Do you have any insight on this?


**Summary Of The Paper:**

The paper considers the problem of learning fair representations via graph neural networks. Concretely, the paper derives an upper bound on the correlation between sensitive attributes and representations and uses this bound to propose various graph augmentations that increase different group fairness metrics (e.g., statistical parity). The proposed augmentations are feature masking, node sampling, and connectivity augmentation. An experimental evaluation on different node classification, link prediction, and edge deletion tasks shows that the proposed method learns representations with higher group fairness while incurring minimal utility decreases.

**Summary Of The Review:**

The relevance of the problem setting, technical and empirical novelty, good presentation, and significant results warrant acceptance of the paper in my opinion.

---

> ### Author Response · Authors · 2021-11-20
> **Response to Reviewer LHQM**
>
> We would like to thank the Reviewer for the supportive and constructive remarks, as well as valuable comments. We have presented our responses to the Reviewer’s questions below:
>
>
> ***
>
> __Q1:__ How do the different graph augmentation strategies interact? The paper conducts an ablation study to compare the performance of different augmentation strategies, but it would be interesting to see if any theoretical conclusions can be made.
>
> __R1:__ We thank the Reviewer for this question. It can be rigorously proved that the proposed edge manipulation schemes only affect the value of $\gamma_2$, but do not affect the value of $\gamma_1$, while node sampling can change both $\gamma_1$ and $\gamma_2$. The feature masking scheme only affects $||\boldsymbol \delta||_1$ but not $\gamma_1$ and $\gamma_2$.
>
> Motivated by this, FairAug is designed to minimize the interaction between different augmentations.
> Specifically, FairAug first employs node sampling, and then uses edge manipulations to reduce $\gamma_2$ without interferencing $\gamma_1$. The feature masking step is applied at last, since it only influences $||\boldsymbol \delta||_1$ but not $\gamma_1$ and $\gamma_2$. This way, we reduce the interference among different schemes. In addition, Appendix A.8 is dedicated to show the sequential and independent effects of the proposed augmentations on the values of $\gamma_1$ and $\gamma_2$. Specifically, Tables 11 and 12 provide the effects of different augmentations on the terms $\gamma_1$ and $\gamma_2$.
>
> Furthermore, to visualize the effects of the augmentations on input graphs, we added Appendix A.9 in the revised manuscript, which demonstrates the influences of the proposed augmentations on toy examples. Specifically, the following guidelines can be provided for the choice of augmentation schemes to obtain better performance:
>
> 1- For input graphs that are highly unbalanced in terms of number of inter-edges ($|\mathcal E^\chi| $) and intra-edges ($|\mathcal E^\omega|$), that is  $|\mathcal E^\omega| \gg |\mathcal E^\chi|$,  the number of edges ($|\mathcal E^\omega|-|\mathcal E^\chi|$) that are needed to be changed is large. Therefore, for such graphs, edge manipulation is not ideal (as also raised by the Reviewer with respect to the ablation study, under Weaknesses), and node sampling is a better choice. This finding can be supported by the results in Table 2, and the  “Edge deletion” part of Appendix A.9.
>
> 2- Node sampling is not effective for graphs where $|\mathcal S^\chi| >  |\mathcal S^\omega|$, and edge deletion is a better choice to reduce bias for such graphs. The results in Table 3 and “Node sampling” part of Appendix A.9 provide support for this claim.
>
>
> ***
>
>
> __Q2:__ The equation for $\gamma_2$  in theorem 1 and section 3.3.3 are inconsistent. Which is the correct one?
>
> __R2:__ We sincerely thank the Reviewer for this keen observation. We have fixed this inconsistency, and used the correct form of $\gamma_2$ (the one presented in Theorem 1) throughout the revised manuscript.
>
>
> ***
>
>
> __Q3:__ Proposition 1 shows that in expectation, the proposed adaptive feature masking scheme results in a lower distribution difference compared to uniform feature masking. However, the uniform feature masking is not the baseline we care about (I think). Instead, we would like to know how the proposed masking scheme affects the distributions compared to no masking at all. Do you have any insight on this?
>
> __R3:__ Thanks for the comment. Compared to no masking at all, it can be rigorously proved that the proposed masking scheme indeed results in a lower distribution difference. Here, the uniform feature masking is used as a baseline, since it also results in a lower distribution difference compared with no masking, and henceforth provides a tighter bound.
>
> In addition, by using uniform feature masking as a baseline, Proposition 1 shows that given a fixed “masking budget”, the proposed adaptive feature masking scheme yields a smaller distribution difference, implying that the adaptive strategy allocates the masking probability in a more effective way than uniform masking.
>
> We have clarified this in the revised paper by adding Footnote 1 in Appendix A.2 as follows:
>
>
>
> “From equation 25, it can be obtained that $ E_{\mathbf q} [||\boldsymbol \delta||]=\sum_{i=1}^F q_i |\delta_i| \leq \sum_{i=1}^F |\delta_i|=||\boldsymbol \delta||_1$, as the probabilities $q_i \in [0,1]$. Here, $||\boldsymbol \delta||_1$ corresponds to the distribution difference with no feature masking ($q_i=1, \forall i$). This implies that no feature masking upper bounds $E[||\boldsymbol \delta||_1]$, which is obtained via a stochastic feature masking.”
>
>
> We hope that our answer satisfactorily addresses the Reviewer’s comment.
>
>
> ***
>
> Overall, we thank the Reviewer again for the comments on our paper, as well as the constructive remarks.

---

### Official Review · Reviewer_gszv · 2021-11-04

**Correctness:** 3
**Technical Novelty And Significance:** 3
**Empirical Novelty And Significance:** 3
**Recommendation:** 6
**Confidence:** 4

**Details Of Ethics Concerns:**

No major concerns.

**Main Review:**

Strengths:

(1) This paper studies an interesting and important problem of GNN fairness.

(2) The proposed method demonstrates effective empirical performance on several benchmark datasets.

(3) The proposed data augmentation tricks are easy to implement and potentially broadly applicable.

Weakness:

(1) The presentation of the analysis can be greatly improved.

    (1.1) The assumptions for Theorem 1 should be stated explicitly in the form of Assumptions. The current writing makes it difficult to check what exactly are being assumed in order to prove Theorem 1.

    (1.2) Some of the assumptions are questionable. For example, it seems that the GNN hidden representations $h_i$'s follow a uniform distribution. How is that possible? And how much does Theorem 1 rely on this assumption?

    (1.3) The intuitions of the analysis should be better explained. For example, $\gamma_2$ takes a very complicated form. What is the intuition of this term?

(2) The connection between the proposed augmentation tricks and the three terms in the upper bound is ambiguous. If the three terms are really important, why don't we directly augment the data by optimizing for those terms? For example, based on the definition of $\gamma_1$, it seems very easy to reduce it to 0 by rewiring the graph. What are the potential trade-offs for not fully optimizing those terms?

(3) In the appendix, it is shown that the proposed tricks can reduce the $\gamma$ values. Could you include an ablation study testing methods that directly optimize for these terms, as mentioned in (2).

(4) Quite a few missing literatures on GNN fairness:

1. A. Bose and W. Hamilton. Compositional fairness constraints for graph embeddings.
2. M. Buyl and T. Bie. The kl-divergence between a graph model and its fair i-projection as a fairness regularizer.
3. Y. Dong, J. Kang, H. Tong, and J. Li. Individual fairness for graph neural networks: a ranking based approach.
4. C. Laclau, I. Redko, M. Choudhary, and C. Largeron. All of the fairness for edge prediction with optimal transport.
5. J. Ma, J. Deng, and Q. Mei. Subgroup generalization and fairness of graph neural networks.
6. Z. Zeng, R. Islam, K. Keya, J. Foulds, Y. Song, and S. Pan. Fair representation learning for heterogeneous information networks.



----------Post Author Response-----------------
I appreciate the author response and revision, which resolved most of my concerns. I therefore raise my evaluation to 6. The proposed method still seems somewhat ad-hoc but overall this work would be an interesting addition to the GNN fairness literature.

**Summary Of The Paper:**

This paper proposes a set of data augmentation approach to improve the fairness of GNNs. It first provides an upper bound on the correlation between the GNN node representations and a given sensitive attribute. Then it proposes three data augmentation tricks to respectively reduce three terms of the upper bound. The proposed method is evaluated on several benchmark datasets to show the effectiveness.

**Summary Of The Review:**

This paper proposes a set of data augmentation tricks that demonstrate empirical gain on GNN fairness. However, the presentation of analysis and motivation is not convincing and can be largely improved.

---

> ### Author Response · Authors · 2021-11-20
> **Response to Reviewer gszv: 4**
>
> __Q6:__  Quite a few missing literatures on GNN fairness:
>
> A. Bose and W. Hamilton. Compositional fairness constraints for graph embeddings.
>
> M. Buyl and T. Bie. The kl-divergence between a graph model and its fair i-projection as a fairness regularizer.
>
> Y. Dong, J. Kang, H. Tong, and J. Li. Individual fairness for graph neural networks: a ranking based approach.
>
> C. Laclau, I. Redko, M. Choudhary, and C. Largeron. All of the fairness for edge prediction with optimal transport.
>
> J. Ma, J. Deng, and Q. Mei. Subgroup generalization and fairness of graph neural networks.
>
> Z. Zeng, R. Islam, K. Keya, J. Foulds, Y. Song, and S. Pan. Fair representation learning for heterogeneous information networks.
>
> __R6:__ We thank the Reviewer for these suggestions. In the revised manuscript, we have included and discussed these references in “Fairness-aware learning on graphs” of Section 2 (Related Work).
>
> ***
>
> We would like to thank the Reviewer again for the valuable comments, and we believe addressing them improved the technical quality, as well as the clarity of presentation of our work.

---

> ### Author Response · Authors · 2021-11-20
> **Response to Reviewer gszv: 2 and 3**
>
> __Q4:__ The connection between the proposed augmentation tricks and the three terms in the upper bound is ambiguous. If the three terms are really important, why don't we directly augment the data by optimizing for those terms? For example, based on the definition of $\gamma_{1}$, it seems very easy to reduce it to 0 by rewiring the graph. What are the potential trade-offs for not fully optimizing those terms?
>
> __R4:__ Thank you for raising this question. We agree that it is easy to reduce $\gamma_{1}$ by rewiring the graph, and it is also possible to directly optimize the terms $\gamma_2$ and $||\boldsymbol{\delta}||_{1}$ as well. The reasons for not directly optimizing to zero is mainly for maintaining utility and stability (see Appendix A.7 for further details).
>
> Note that the proposed node sampling scheme in Algorithm 1 is designed to reduce $\gamma_{1}$ to 0 assuming $|\mathcal{S}^{\omega}| > |\mathcal{S}^{\chi}|$. However, for graph structures where there is a significant unbalance in terms of $\mathcal{S}^{\chi}$ and $\mathcal{S}^{\omega}$ (e.g., Facebook networks in the experiments), a node sampling strategy that ensures $\gamma_{1}=0$ would result in the deletion of a large portion of nodes and henceforth edges. This may affect the utility and stability of the trained model. Hence, we limit the proportion of nodes that can be removed in the sampling process (as explained in Appendix A.7). This is the key reason for $\gamma_{1}$ values not directly becoming 0 after the application of node sampling in Table 12.
>
> For $||\boldsymbol \delta||_1$, masking all features where $\boldsymbol \mu_0-\boldsymbol \mu_1$ is nonzero results in $||\boldsymbol \delta||_1=0$, which however leads to the loss of all nodal features in most of the scenarios. Such masking prevents the learning of nodal representations, as feature aggregation will generate all zero vectors for all nodes, and is not effective even though it is indeed fair to the maximum extent. Thus, we employ a hyperparameter $\alpha$ to specify the amount of nodal features that will be masked and the same amount of nodal features are masked in all baselines and our work. The design idea of the proposed feature masking scheme is to reduce $||\boldsymbol \delta||_1$ in an effective way for a fixed masking budget, which is also demonstrated by Proposition 1.
>
> For $\gamma_{2}$, direct optimization of it requires the consideration of each node independently, as the number of neighbors from different sensitive groups should be equalized for each node. Such a node-level scheme leads to high complexity for large scale graphs. This motivates us to resort to the more efficient global schemes instead. Meanwhile, direct optimization of $\gamma_{2}$ may also influence the value of $\gamma_{1}$, since deletion/addition of inter-edges for each node can affect the number of nodes in $\mathcal{S}^{\chi}$. Global edge deletion framework employed in the present paper does not delete any of the inter-edges for $\pi=1$, hence can prevent such an interference. Similarly, the global edge addition is designed to avoid interference with $\gamma_{1}$ by only creating edges between the nodes in $\mathcal{S}^{\chi}$.
>
>
> ***
>
>
> __Q5:__ In the appendix, it is shown that the proposed tricks can reduce the γ values. Could you include an ablation study testing methods that directly optimize for these terms, as mentioned in (2)（Now Q4）
>
> __R5:__ We would like to thank the Reviewer for this valuable suggestion. Following this suggestion, we included the individual/sequential effects of optimal graph topology augmentations on $\gamma_{1}$ and $\gamma_{2}$ in Appendix A.10, specifically in Tables 13-16.
> As optimal feature masking would result in masking all features (please see our response to Q4), we did not consider that case. For the optimal node sampling, the method in Algorithm 1 is employed without any limit on the sampling budget, while the optimal edge manipulations work on each node independently to equalize the number of neighbors from different sensitive groups. We demonstrated that these schemes can indeed make $\gamma_{1}$ (node sampling) and $\gamma_{2}$ (edge manipulations) zero. However, the results also show the inference of such edge manipulations on the value of $\gamma_{1}$, which is consistent with our response to Comment 4.

---

> ### Author Response · Authors · 2021-11-20
> **Response to Reviewer gszv: 1.3**
>
> __Q3:__ The intuitions of the analysis should be better explained. For example, γ2 takes a very complicated form. What is the intuition of this term?
>
> __R3:__ Thank you for your question. The goal of the analysis is two-fold:
>
>  i) Theoretically explaining the sources of bias via identifying the key factors ($\boldsymbol \delta$, $\gamma_1$, $\gamma_2$) that may contribute to the correlation between sensitive attributes __s__ and aggregated representations __Z__, as node features that are correlated with sensitive attributes lead to bias [R1],
>
> ii) Providing guidelines for the design of ensuing fairness-aware schemes in order to reduce the corresponding terms ($\boldsymbol \delta, \gamma_1, \gamma_2$).
> Specifically, the intuition of the related terms are as follows:
>
> $\boldsymbol \delta :=\boldsymbol \mu_{0}-\boldsymbol \mu_{1}$: $\boldsymbol \delta$ indicates that the discrepancy between the average input feature vectors of the two sensitive groups may lead to bias in representations. This intuition in turn provides a guideline for the design of our feature masking scheme, which masks the features that have larger discrepancy ($\boldsymbol \delta$) with higher probabilities ($\mathbf p^{(m)}$).
>
> $\gamma_1:=\big|1-\frac{|\mathcal S^\chi_0|}{|\mathcal S_0|}-\frac{|\mathcal S^\chi_1|}{|S_{1}|} \big|$: It can be observed that $\gamma_1$ is minimized when $|\mathcal S_0^\omega|=|\mathcal S_0^\chi|$ and $|\mathcal S_1^\omega|=|\mathcal S_1^\chi|$. Here, $\mathcal S_i^\omega$ denotes the set of nodes from sensitive group i with only intra-edges, and $\mathcal S_i^\chi$ stands for the set of nodes from sensitive group i with at least one inter-edge. Therefore, decreasing $\gamma_1$ suggests a more balanced graph population in terms of the cardinality of $\mathcal S^\chi$ and $\mathcal S^\omega$. This intuition serves as a guideline for our node sampling scheme in Section 3.3.2.
>
> Thanks to the Reviewer’s suggestion, we added the following text to Section 3.3.2 to clarify the intuition of $\gamma_1$ in the analysis:
>
> “A small $\gamma_1$ suggests a more balanced population distribution with respect to $|\mathcal S^\chi|$ and $|\mathcal S^\omega|$.”.
>
>
>  $\gamma_2:= |1-2\min(mean(\frac{d_m^\chi}{d_m^\chi+d_m^\omega}| v_m \in \mathcal S_0),mean(\frac{d_n^\chi}{d_n^\chi+d_n^\omega}| v_n \in \mathcal S_1))|$:  $\gamma_2$ is derived from equation 19 in Appendix A.1. As mentioned in Section 3.3.3, it can be observed that $\gamma_2=0$, if all nodes in the network have the same number of neighbors from each sensitive group, i.e., $d_i^\omega=d_i^\chi, \forall v_i \in \mathcal V$. Therefore, this term suggests that the number of neighbors from different sensitive groups should be balanced to reduce the bias in information aggregation. This intuition motivates our edge augmentation schemes in Section 3.3.3. It is worth mentioning that this finding also coincides with and henceforth provides potential theoretical support for the design of Fairwalk [R2] which aims to equalize the selection probabilities of each sensitive group in a random walk while creating random walk-based node representations.
>
>
> To clarify the intuition behind $\gamma_2$, the following text is added to Section 3.3.3 in the revised manuscript:
>
> “Note that this finding is parallel to the main design idea of Fairwalk [R2] in which the transition probabilities are equalized for different sensitive groups in random walks in order to reduce bias in random walk-based representations.”
>
> [R1] A methodology for direct and indirect discrimination prevention in data mining, TKDE 2013
>
> [R2] Fairwalk: Towards Fair Graph Embedding, IJCAI, 2019

---

> ### Author Response · Authors · 2021-11-20
> **Response to Reviewer gszv: 1.1 and 1.2**
>
> We thank the Reviewer for the insightful comments regarding our work. We have addressed the Reviewer’s comments, and placed our responses to each comment below.
>
> ***
>
> __Q1:__ The assumptions for Theorem 1 should be stated explicitly in the form of Assumptions. The current writing makes it difficult to check what exactly are being assumed in order to prove Theorem 1.
>
> __R1:__ Following the Reviewer’s suggestions, we revised Section 3.2 by explicitly stating the assumptions utilized in the analysis item by item. In the revised version, the assumptions are listed as follows:
>
> “In the analysis, following assumptions are made:
>
> __A1:__ Node representations have sample means $\boldsymbol \mu_{0}$ and $\boldsymbol \mu_{1}$ respectively across each group, where $\boldsymbol \mu_{i} =mean(\mathbf h_j \mid v_{j} \in \mathcal S_{i})$. Throughout the paper, $mean(\cdot)$ denotes the sample mean operation.
>
> __A2:__  Node representations have finite maximal deviations $\Delta_0$ and $\Delta_1$: That is, $\left\|\mathbf h_j-\boldsymbol \mu_{i}\right\|_\infty \leq \Delta_i$, $\forall v_j \in S_i$ with $i \in$ {0,1}."
>
> We would like to thank the Reviewer for this suggestion, and we hope the revised manuscript has become more clear.
>
> ***
>
>
> __Q2:__ Some of the assumptions are questionable. For example, it seems that the GNN hidden representations hi's follow a uniform distribution. How is that possible? And how much does Theorem 1 rely on this assumption?
>
> __R2:__ Thanks to this comment, we thoroughly examined our analysis. and found that the bound in Theorem 1 holds without the uniform distribution assumption. Hence, we have revised our manuscript and removed this assumption accordingly, see also our response to Comment 1 where the assumptions are explicitly stated as suggested. We sincerely thank the Reviewer for this keen observation and insightful comment.

---

### Author Response · Authors · 2021-11-20
**Revised Manuscript and Responses to Reviewers**

We would like to thank the Reviewers for their detailed reviews and constructive suggestions. We have addressed the questions raised by the Reviewers, and presented the detailed responses in a point-to-point manner. In summary, according to the comments of the Reviewers, we have revised our manuscript as follows:

1- We carried out sensitivity analyses for hyperparameters by adding more experiments in the revised Appendix A.7, see Tables 7-10.

2- In order to show the effects of the proposed augmentations in a more intuitive manner, we introduced Appendix A.9, where we illustratively presented the working mechanisms of the proposed schemes (Figures 2-4), on two toy example graphs (Figure 1).

3- We have added Appendix A.10, where we examined the effects of the “optimal” graph topology augmentation schemes that exactly minimize the terms in Theorem 1’s upper bound ($\gamma_1$ via node sampling, $\gamma_2$ via edge manipulations), see Tables 13-16.

4- We have revised the “Fairness-aware learning on graphs” part of Section 2 (Related Work), in order to further clarify and highlight the novelty and advantages of our work compared with prior art.

5- We relaxed and explicitly stated the assumptions of Theorem 1.

6- Following the Reviewers’ suggestions, other minor revisions were also carried out to improve the clarity of presentation.

For more details, please check our point-to-point responses below, as well as the uploaded revised manuscript. All changes in the revised manuscript are highlighted in purple color, while the sentences from the original submission are kept in black color.

---

### Decision · Program_Chairs · 2022-01-20

**Decision:**

Reject

**Comment:**

This paper studies augmentation-based methods to improve GNN fairness.
Specifically, based on upper bounds, they propose augmentation tricks to reduce such bounds, empirically validated from benchmark datasets.

Before rebuttal, there was a negative consensus that evaluation results are inconclusive and important state-of-the-arts are not discussed.
Authors have significantly revised to address the concerns of some reviewers (gszb and LHQM), though some concerns still remain that the scheme is rather ad-hoc.
Meanwhile, reviewer xMz4 did not find rebuttal sufficient, as some valid comments were not fully discussed.
First, datasets where sensitive attributes are the inherent attributes of instances are more suitable for evaluation, which has not been properly addressed by authors.
Second, important baselines were mentioned by xMz4, but they were only mentioned briefly in the new section of related work in the revised work.
For example, (Agarwal et al 2021) is mentioned as dealing with counterfactual fairness only, but statistic parity studied in this paper is highly related to this concept.
Similarly, this work can be viewed as an ad-hoc extension for fairness of GCA, without in-depth discussions to compare/contrast with these work, and to better highlight novelty/distinction.

Summing up reviewer discussions, we conclude this paper is not ready yet as an ICLR publication.

---

> ### Public Comment · ~Oyku_Deniz_Kose1 · 2022-01-29
> **Response to Paper Decision**
>
> We would like to thank the Area Chair and Reviewers for handling our submission.
>
> ```
> Authors have significantly revised to address the concerns of some reviewers (gszb and LHQM), though some concerns still remain that the scheme is rather ad-hoc.
> ```
>
> Our designs are based on rigorous theoretical analysis quantifying the sources of bias for a general GNN framework. Specifically, the correlation between sensitive attributes and aggregated representations generated by a GNN layer is quantitatively analyzed, which motivates several frameworks to reduce the bias from the source. Resulting schemes are henceforth well-justified by rigorous analysis, and are applicable to several GNN-based learning frameworks (two of which discussed within the paper).
>
> ```
> Meanwhile, reviewer xMz4 did not find rebuttal sufficient, as some valid comments were not fully discussed.
> First, datasets where sensitive attributes are the inherent attributes of instances are more suitable for evaluation, which has not been properly addressed by authors.
> ```
>
> While we did not hear any feedback in the rebuttal procedure from reviewer xMz4, we tried our best to clarify the misunderstanding regarding the use of datasets. As we also mentioned in our response to reviewer xMz4, __our paper is not the first study utilizing datasets where sensitive attributes are not the inherent attributes, [R1,R2] utilize exactly the same experimental setup.__ Such a setup is utilized due to the lack of datasets with actual sensitive attributes, and the fact that the fundamental learning problem is decoupled from its societal implication.
>
> [R1] On Dyadic Fairness: Exploring and Mitigating Bias in Graph Connections, ICLR, 2021
>
> [R2] Biased Edge Dropout for Enhancing Fairness in Graph Representation Learning, IEEE Transactions on Artificial Intelligence, 2021
>
> ```
> Second, important baselines were mentioned by xMz4, but they were only mentioned briefly in the new section of related work in the revised work. For example, (Agarwal et al 2021) is mentioned as dealing with counterfactual fairness only, but statistic parity studied in this paper is highly related to this concept.
> ```
>
> There must be some misunderstanding. __We compared our schemes with (Agarwal et al 2021, NIFTY) in our experiments both in the initial and revised submissions.__ Given dozens of cited related works, one can easily find references we did not compare to. But we cited them and clarified the novelty of our work. Overall, __we compared to 7 baselines including the most recent publications.__
>
> ```
> Similarly, this work can be viewed as an ad-hoc extension for fairness of GCA, without in-depth discussions to compare/contrast with these work, and to better highlight novelty/distinction.
> ```
>
> We respectfully disagree that  “this work can be viewed as an ad-hoc extension for fairness of GCA”. The differences and novelty are:
>
> 1. We provided theoretical analysis for bias in the general GNN framework, which is not limited to GCA.
>
> 2. Our experimental results in Table 4 demonstrate that the efficacy of our framework is not limited to contrastive learning. Hence, the present work is not an extension of GCA (Graph Contrastive Learning with Adaptive Augmentation), which specifically focuses on contrastive learning.